# Diurnal Cycle in Surface Incident Solar Radiation Characterized by CERES Satellite Retrieval

## Lu Lu and Qian Ma *

State Key Laboratory of Earth Surface Processes and Resource Ecology, College of Global Change and Earth System Science, Faculty of Geographical Science, Beijing Normal University, Beijing 100875, China; 202121490006@mail.bnu.edu.cn
* Correspondence: maqian@bnu.edu.cn

**Abstract:** Surface incident solar radiation ($R_s$) plays an important role in climate change on Earth. Recently, the use of satellite-retrieved datasets to obtain global-scale $R_s$ with high spatial and temporal resolutions has become an indispensable tool for research in related fields. Many studies were carried out for $R_s$ evaluation based on the monthly satellite retrievals; however, few evaluations have been performed on their diurnal variation in $R_s$. This study used independently widely distributed ground-based data from the Baseline Surface Radiation Network (BSRN) to evaluate hourly $R_s$ from the Clouds and the Earth's Radiant Energy System Synoptic (CERES) SYN1deg–1Hour product through a detrended standardization process. Furthermore, we explored the influence of cloud cover and aerosols on the diurnal variation in $R_s$. We found that CERES-retrieved $R_s$ performs better at midday than at 7:00–9:00 and 15:00–17:00. For spatial distribution, CERES-retrieved $R_s$ performs better over the continent than over the island/coast and polar regions. The Bias, MAB and RMSE in CERES-retrieved $R_s$ under clear-sky conditions are rather small, although the correlation coefficients are slightly lower than those under overcast-sky conditions from 9:00 to 15:00. In addition, the range in $R_s$ bias caused by cloud cover is 1.97–5.38%, which is significantly larger than 0.31–2.52% by AOD.

**Keywords:** BSRN; CERES; solar radiation; diurnal variation





## 1. Introduction

Surface incident solar radiation ($R_s$) plays an important role in climate change as it is the major energy source in the Earth system [1,2]. The performance of $R_s$ is closely related to water cycle, as it significantly affects the evaporation of surface water [3]. Robock et al. [4] have shown that the changes in soil temperature are consistent with those in $R_s$ during the global brightening and dimming period. Changes in $R_s$ affect the melting and growth of glaciers as well [5]; for example, the snow cover in the northern hemisphere did not change significantly before the 1980s but exhibited a sharp downward trend in the brightening period after the 1980s [6]. Several studies suggest that $R_s$ also has an impact on ecosystems, driving crop growth as an energy factor [7,8]. In addition, the spatial and temporal distribution of $R_s$ is the basis for energy policy decision making on solar power [9]. Therefore, decreases and increases in $R_s$ (also known as global dimming and brightening, respectively) have received widespread attention [10–12]. The variations in cloud cover, aerosols, and other factors were also explored, as they contributed more to the change in $R_s$ [13,14]. Cloud cover regulates the radiative energy balance of the atmosphere [15]. Compared with a clear-sky atmosphere, cloud cover can absorb and reflect a large amount of incident shortwave radiation, which has a cooling effect on the subsurface. Aerosols can absorb and scatter solar radiation, which can block incident solar radiation, especially by reducing the passage of ultraviolet rays, weakening the solar radiation reaching the ground [16].

The diurnal cycle in $R_s$ due to the Earth's rotation can cause diurnal changes in the surface and atmospheric state in all regions except the polar regions (where solar variability is more seasonal) through a variety of physical processes [17]. Ye et al. [18] noted that the diurnal temperature range (DTR) decreased rapidly in China from the 1960s to the 1980s when it became significantly dimmer, while in the early 1990s, the decline in DTR stopped during the period when $R_s$ changed from dimming to brightening, which was confirmed by Du et al. [19]. Both sea surface temperature variability in the western Pacific warm pool [20] and the diurnal behavior of the spiral rainbands [21] are modulated by the diurnal cycle in $R_s$.

Currently, $R_s$ can be obtained from ground-based observations, reanalyses datasets, and satellite-retrieved datasets [22–24]. The ground-based observations of $R_s$ have the highest accuracy but are sparsely distributed [9]. Reanalyses datasets have better spatial continuity, but the bias in cloud and aerosol simulations can lead to low accuracy in $R_s$ [25,26]. The accuracy of the satellite retrieved data is relatively higher than that of reanalyses, as clouds and aerosols observed by satellite were used in the radiative transfer model [27].

Most studies have evaluated satellite-retrieved $R_s$ on monthly and annual time scales. These studies have shown that satellite-retrieved datasets are less biased and have better accuracy than reanalyses datasets at long time scales [14,28,29]. However, few evaluations have been performed on satellite-retrieved $R_s$ on the diurnal scale. Therefore, a comprehensive and detailed evaluation of satellite-retrieved $R_s$ on an hourly scale is needed. In this study, we compared Clouds and the Earth's Radiant Energy System Synoptic (CERES)-retrieved hourly $R_s$ with Baseline Solar Radiation Network (BSRN) observations to quantify their differences in the diurnal cycle of $R_s$. Furthermore, the influence of clouds and aerosols on the diurnal variation in $R_s$ was explored.

## 2. Data and Methods

### 2.1. Ground-Based Observation Data

Ground-based observation data with high-quality instrumentation and long-term maintenance provide the most reliable and accurate $R_s$ [9]. In 1988, the WMO proposed the establishment of a new international ground-based radiation baseline network (i.e., BSRN) under the World Climate Research Program (WCRP) [30]. BSRN provides high-temporal resolution ground-based radiation observations (1 min) that can be used to validate satellite-retrieved datasets, improve radiative transfer calculations in climate models, and support the detection and monitoring of long-term changes in ground-based radiation fluxes [31].

In this study, we use ground-based observations from the BSRN to evaluate the satellite-retrieved dataset. As CERES data began in March 2000, the research period in this study is March 2000–July 2021. Seventy–three stations observed $R_s$ during this period, as shown in Figure 1. Figure 2 shows the length of the record at each site. Twenty sites marked in red were excluded because their recorded periods were less than 4 years. Finally, 53 sites marked in blue color were used for evaluation in this work. Of these, 38 are located on the continent, 8 are located on the continent along the island/coast, and 8 are located on the continent in the polar region.

Thermophile radiometers suffer a negative bias because of the infrared loss to the sky at nighttime [32]. Consequently, measurement data at nighttime were abandoned in this study. We focus on the period from 7:00 to 17:00 each day, as the sample size of the data in this period is larger than 200,000 (see Figure 3), accounting for more than 50% of the data for each hour.

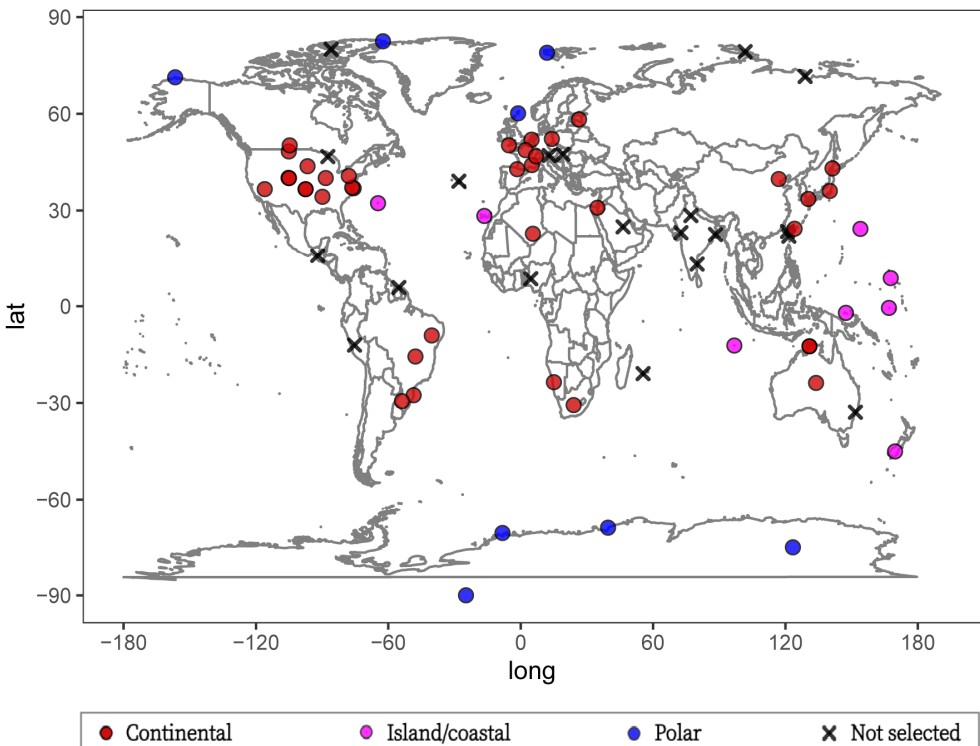

**Figure 1.** Geographical distribution of observation sites used for the evaluation of satellite-retrieved $R_s$ data. The blue, red, and magenta circles indicate the site location in the polar (Arctic or Antarctic), continent, and island/coast, respectively. The black crosses indicate the sites excluded in this study as fewer data were available.

### 2.2. Satellite-Retrieved Dataset

Clouds and the Earth's Radiant Energy System (CERES) is an investigation to provide earth radiation budget data through satellite sensing, which is produced, archived, and made available to the scientific community by the Atmospheric Sciences Data Center (ASDC), the Langley Research Center (LaRC), and the National Aeronautics and Space Administration (NASA) [33].

The CERES SYN Ed4.1 product contains hourly $R_s$, cloud cover, and aerosol optical depths with a $1° \times 1°$ spatial resolution. Cloud properties from high-resolution imagers on several satellites were precisely matched with broadband radiance data. CERES has been providing clouds since 2000 using the algorithm developed for the second edition of CERES (Ed2) until 2002. To improve the accuracy of the clouds, CERES Edition 4 (Ed4) applies the revised algorithm [34]. The aerosols are from the Multi-scale Atmospheric Transport and Chemistry (MATCH) model constituents and the NASA-GSFC Moderate-resolution Imaging Spectroradiometer (MODIS) MOD04_L2/MYD04_L2 products [35]. The clouds and aerosols are used as inputs of the Fu–Liou radiative transfer model. Additional inputs are pressure, temperature and water vapor profiles from the Global Modeling and Assimilation Office (GMAO) Goddard Earth Observing System Model (GEOS). Gaseous absorption in the shortwave region is treated by the method described in Kato et al. [36], and absorption by water vapor, carbon dioxide, ozone, methane and oxygen were considered. Then, the adjusted fluxes were derived by constraining the calculated at the top of the atmosphere (TOA) fluxes to the observed CERES TOA fluxes. $R_s$ in the CERES-SYN Ed4.1 products were computed hourly in approximate equal-area grid boxes [37].

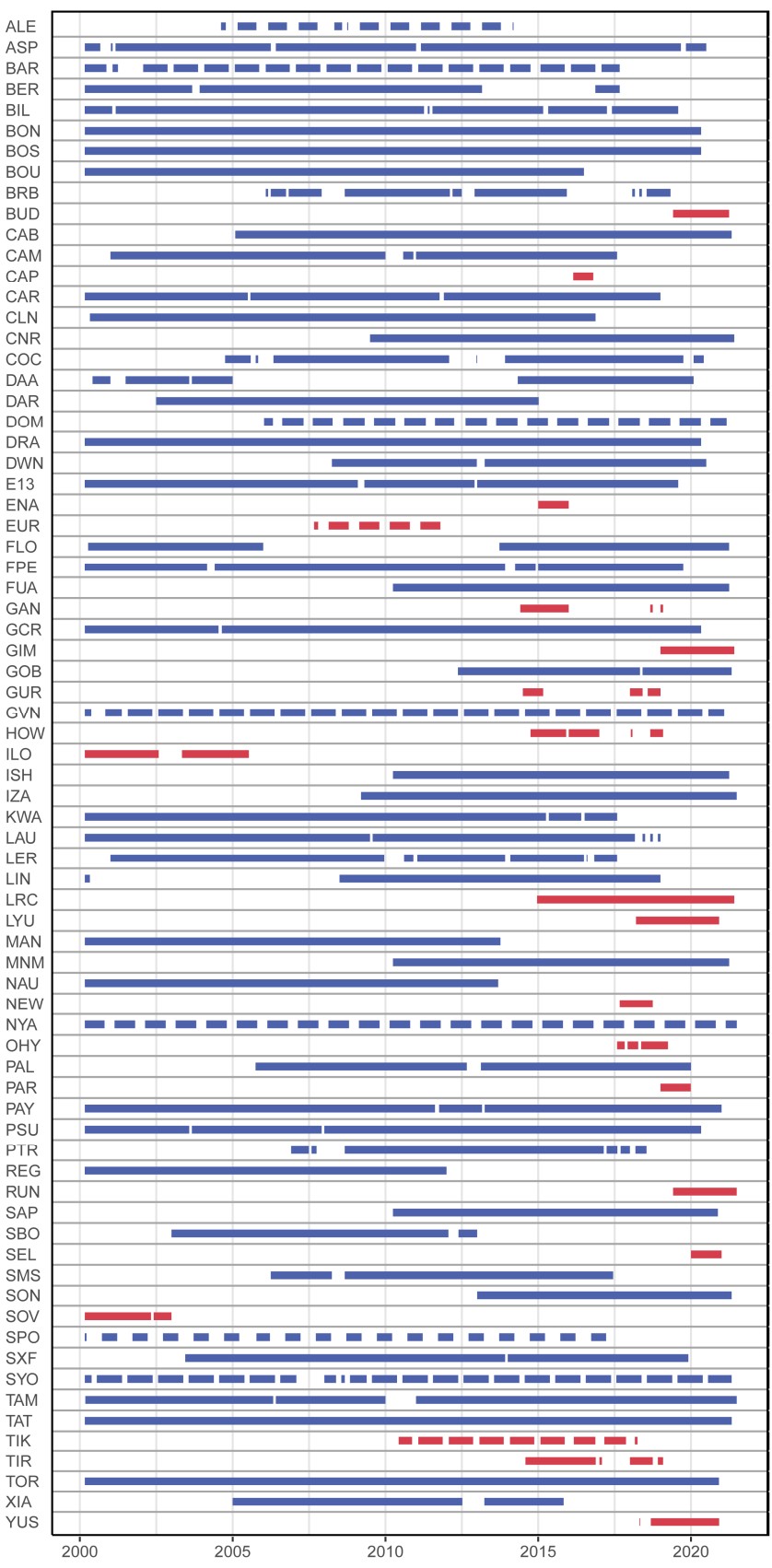

**Figure 2.** Time duration of each site in BSRN for 2000–2021. Red indicates that sites with short measurement periods are excluded from this work; blue indicates that sites are used in this work.

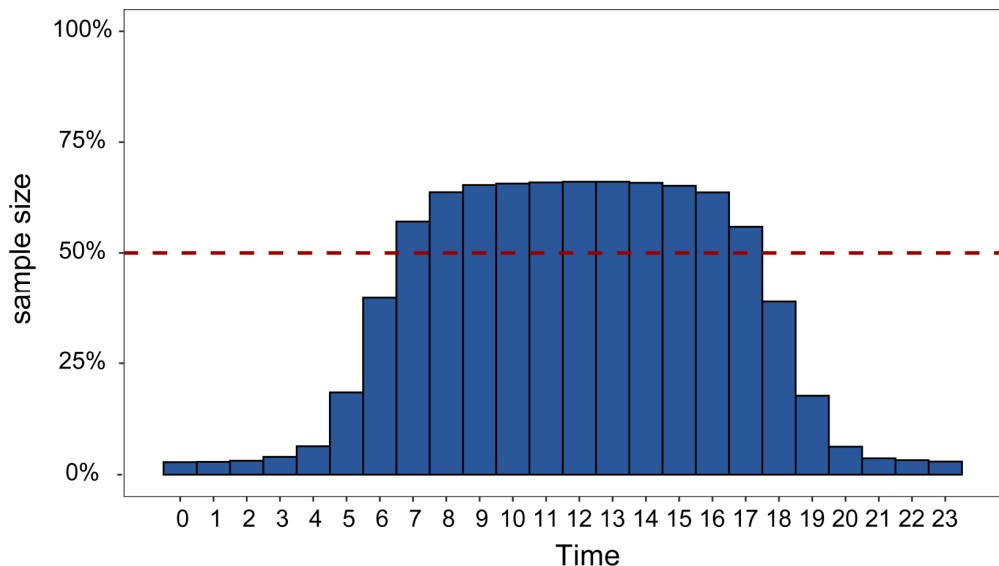

**Figure 3.** The sample size of the observation data at different times. The red line indicates 50% of the total sample size.

*2.3. Methods*

Taking into account that the $R_s$ measurement stations are distributed throughout the Earth, we converted UTC to local time for subsequent evaluation. To reduce the diurnal cycle and seasonal cycle in the $R_s$ on the evaluation results, the observational data and CERES retrieved data were standardized as follows:

$$SW_{i,std} = \frac{SW_i}{SW_{i,TOA}} \tag{1}$$

where $SW_{i,std}$ represents the standardized radiation value at hour $i$, $SW_i$ represents the original radiation value at hour $i$ and $SW_{i,TOA}$ represents the radiation value at the top of the atmosphere at hour $i$. Solar radiation at the top of atmosphere ($SW_{TOA}$) is affected by the solar zenith angle and Sun–Earth distance, so $SW_{TOA}$ contains diurnal and seasonal variation, as shown in Figure 4. Both the observational $R_s$ data and CERES retrieved $R_s$ data were standardized by dividing $SW_{TOA}$ from CERES. The standardized $R_s$ data are greater than or equal to 0 and less than 1 and unitless. It is worth noting that the $R_s$ appearing in the following is the standardized one.

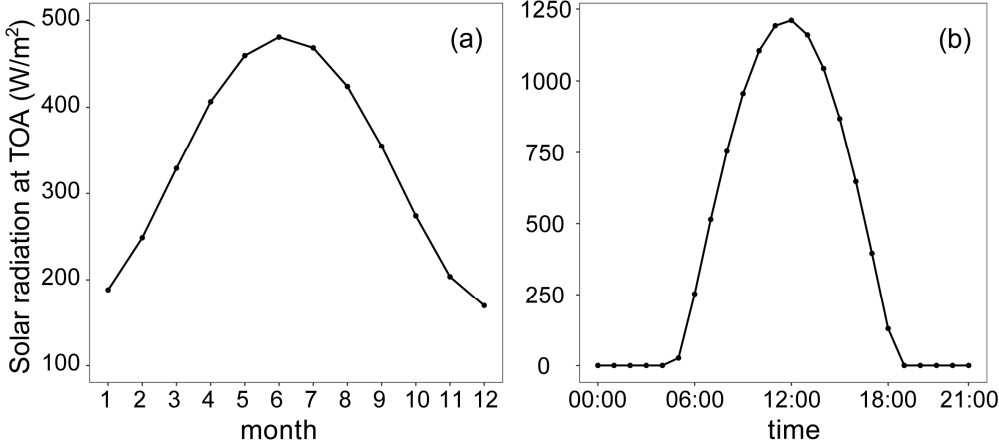

**Figure 4.** (**a**) Monthly solar radiation at TOA at XIA station in 2005; (**b**) hourly solar radiation at TOA at XIA station on 25 April 2005.

In this study, the Bias, mean absolute bias (MAB), root-mean-square error (RMSE), and correlation coefficient (R) are used to evaluate satellite radiation data with ground-based observations, as shown in Equations (2)–(5).

$$\text{Bias} = \frac{1}{n}\sum_{i=1}^{n}(S_i - O_i) \tag{2}$$

$$\text{MAB} = \frac{1}{n}\sum_{i=1}^{n}|S_i - O_i| \tag{3}$$

$$\text{RMSE} = \sqrt{\frac{\sum_{i=1}^{n}(S_i - O_i)^2}{n}} \tag{4}$$

$$\text{R} = \frac{\sum_{i=1}^{n}(S_i - \overline{S})(O_i - \overline{O})}{\sum_{i=1}^{n}(S_i - \overline{S})^2 \sum_{i=1}^{n}(O_i - \overline{O})^2} \tag{5}$$

where $S_i$ is the satellite-retrieved $R_s$ at hour $i$ and $O_i$ is the observed $R_s$ at hour $i$. The mean absolute bias was used here to avoid the offsetting of positive and negative deviations.

## 3. Results

### 3.1. Difference between CERES-Retrieved and BSRN Hourly $R_s$

Most $R_s$ data are in the range from 0.50 to 0.75, as shown in Figure 5l. Hourly $R_s$ were underestimated by −1.10% in CERES. The MAB between satellite-retrieved and observed $R_s$ is less than 10%, while the RMSE is 12.88% for the whole period. As the seasonal and diurnal cycles were removed, their correlation coefficient was 0.83, which is much less than 0.92 for calculation with absolute values. This is consistent with the value of 0.95 noted by David et al. [35].

Evaluations were also conducted each hour, as shown in Figure 5a–k. Hourly $R_s$ retrieved by CERES show negative Bias from 7:00 to17:00, with the largest Bias of −2.29% at 7:00 and the smallest Bias of −0.43% at 13:00. The MAB ranges from 8.77% to 8.92% for 10:00–13:00 and is relatively large, 10.03% and 9.89%, for 7:00 and 17:00, respectively. For other hours, MAB ranges from 9.09% to 9.52%. It decreases until midday and increases afterward. The RMSE is smallest at 10:00 (12.51%) and largest at 7:00 (13.66%). The diurnal variation of RMSE is similar to that of MAB. R is higher than 0.8 most of the time, except at 7:00 and 17:00. R is higher at midday and lower at 7:00 and 17:00. These results suggest that the CERES-retrieved $R_s$ performs better at midday than at 7:00–8:00 and 16:00–17:00. The smaller difference between satellite-retrieved and observed $R_s$ from 10:00 to 13:00 may be attributed to the orbit overpass times related to the two sensors, Terra and Aqua. Terra is in a descending sun-synchronous orbit with an equator-crossing time at 10:30, while Aqua is in ascending sun-synchronous orbits with an equator-crossing time at 13:30 [38].

Most of the stations show negative Bias, especially at 7:00 and 17:00, and it is particularly evident for stations located along the island/coast (Figure 6 and Table A1). MAB and RMSE are especially larger at 7:00 and 17:00 with relatively smaller R for most stations, as shown in Figures 7–9 and Table A1. They perform better at midday. These are consistent with those shown in Figure 6. Notably, the largest Bias (−21.95% for Bias, 22.93% for MAB, 25.79% for RMSE, and 0.56 for R) was found at the IZA station located in Tenerife (Canary Islands, Spain) [39], which is consistent with the results of Hao et al. [40] and Tang et al. [22]. This may be attributed to its high altitude above the subtropical inversion layer, as cloud cover only affects the lower part of the area (below 2000 m), while the upper part of the island is cloudless [41].

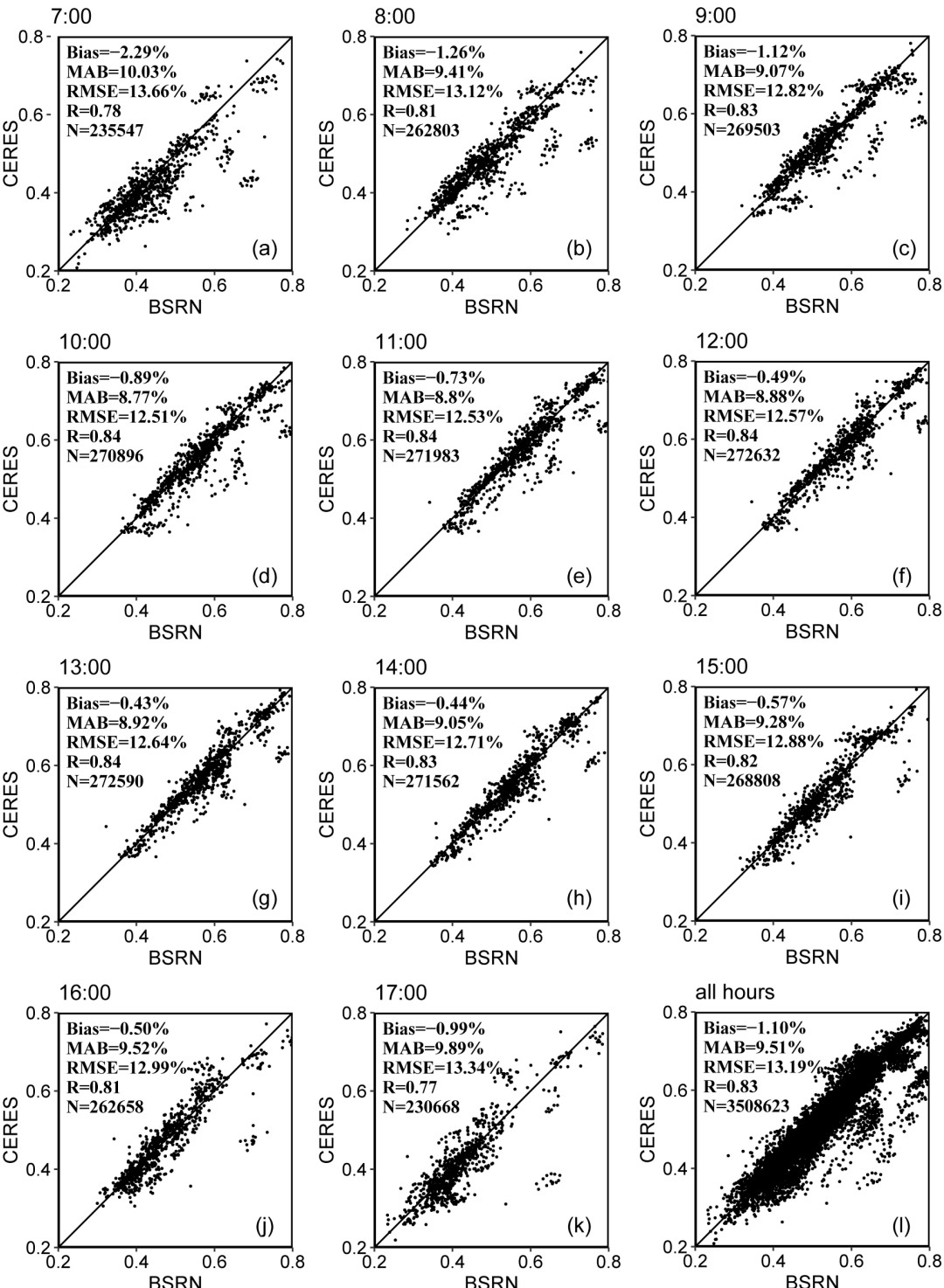

**Figure 5.** Scatter plots of annual average of hourly CERES retrieved and observed $R_s$ from 2000 to 2021 for each hour from 7:00 to 17:00 and all hours. The statistical parameters in the upper left corner of (**a**–**l**) were calculated using $R_s$ at 53 stations in Figure 1, and N is sample hours participating in the calculation.

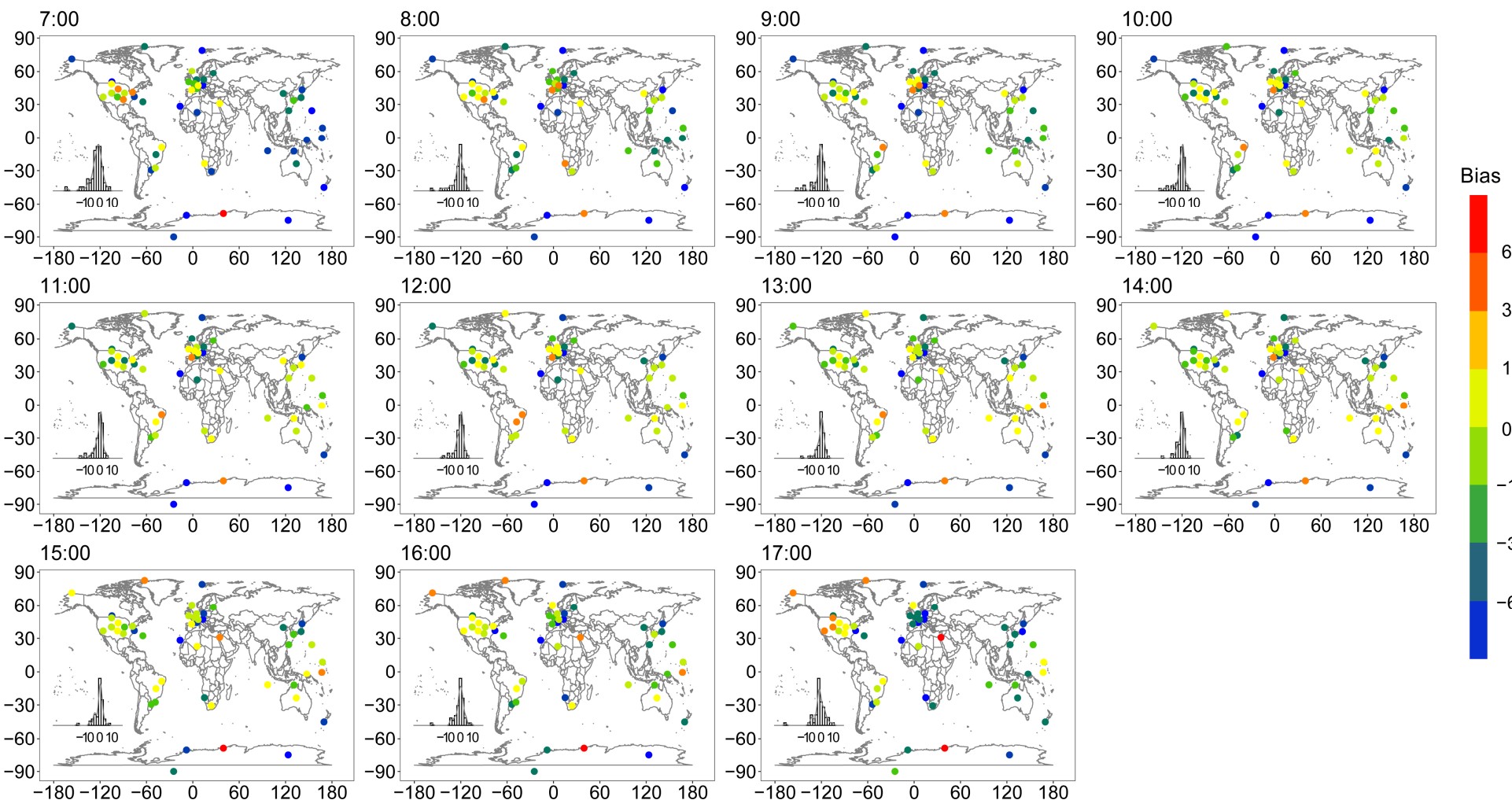

**Figure 6.** Bias between hourly CERES-retrieved $R_s$ and observed $R_s$ in each station for individual hours from 7:00 to 17:00. Unit: %.

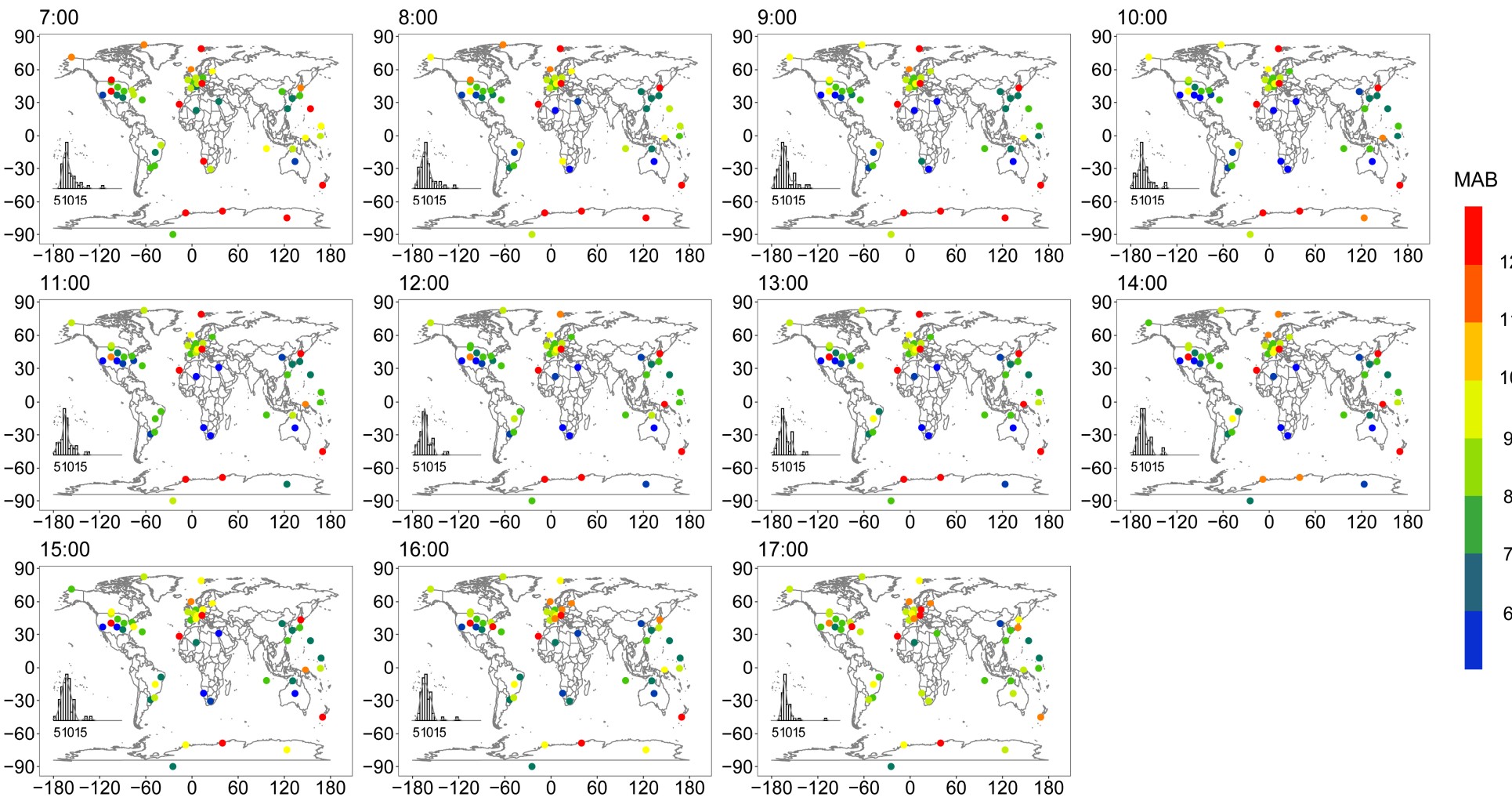

**Figure 7.** Mean absolute bias (MAB) between hourly CERES-retrieved $R_s$ and observed $R_s$ in each station for individual hours from 7:00 to 17:00. Unit: %.

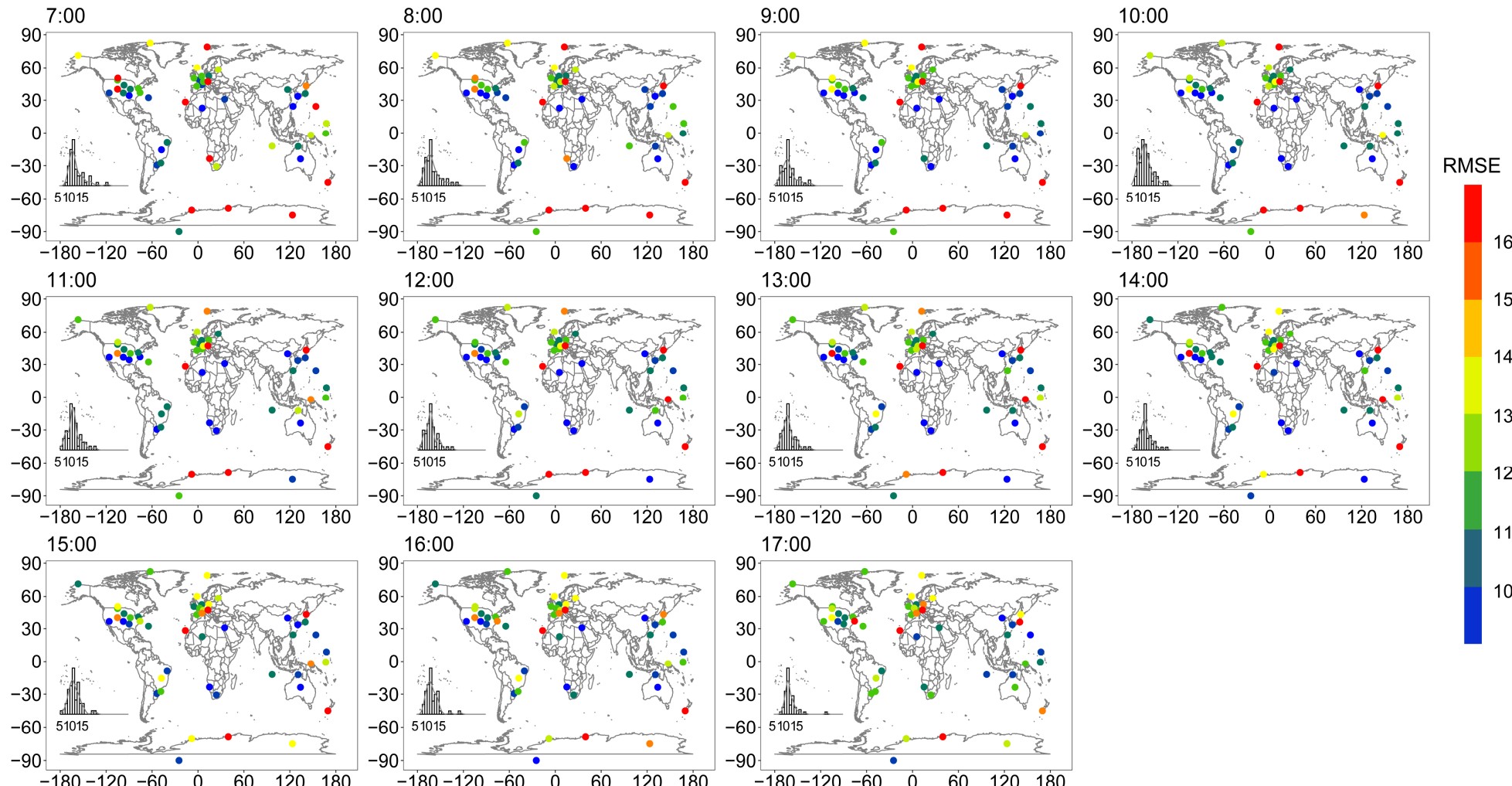

**Figure 8.** Root-mean-square error (RMSE) between hourly CERES-retrieved $R_s$ and observed $R_s$ in each station for individual hours from 7:00 to 17:00. Unit: %.

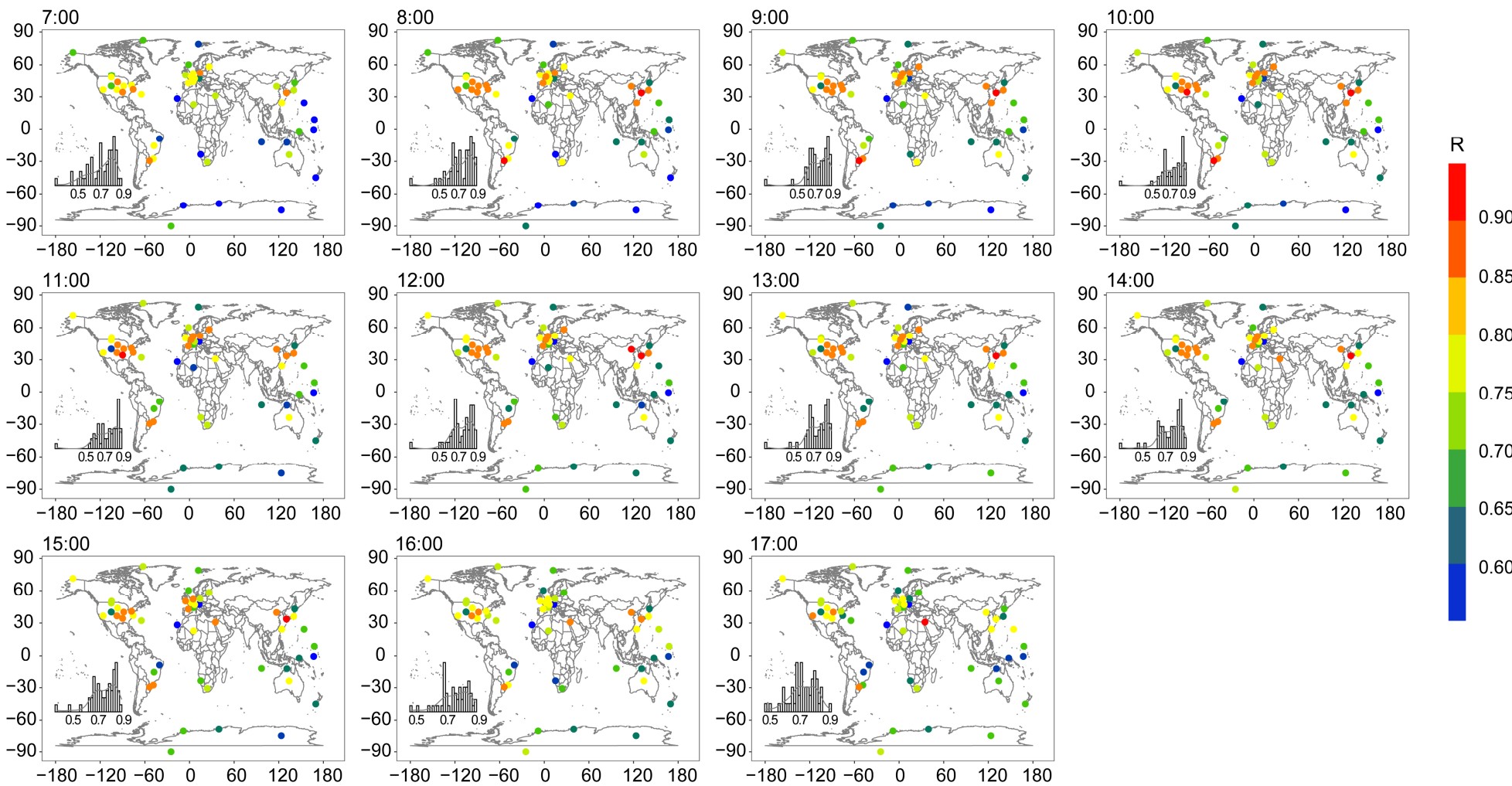

**Figure 9.** Correlation coefficients (R) between hourly CERES-retrieved $R_s$ and observed $R_s$ in each station for individual hours from 7:00 to 17:00.

Figure 10 and Table 1 summarize the statistical parameters for different types of stations. The negative Bias for the continental stations ($-0.01$–$-1.38$%) is smaller than that for the island/coastal stations ($-1.84$–$-8.16$%), and they are both underestimated more at 7:00 and 17:00 and less at midday. The stations located in the polar regions show the opposite variation, with negative Bias increasing until 9:00, then decreasing throughout the day and finally showing a positive Bias after 16:00. The MAB for continental stations ranges from 8.14% to 9.66%, which is smaller than that for island/coastal (10.67–13.54%) and polar (9.84–12.11%) stations. The RMSE for continental stations ranges from 11.44% to 12.51%, which again is smaller than that for island/coastal (13.94–16.93%) and polar (13.13–15.74%) stations. The R value for continental stations ranges from 0.74 to 0.80, which is higher than that for island/coastal (0.59–0.70) and polar (0.66–0.76) stations. The CERES-retrieved $R_s$ performs better at most continental stations, although the spatial variabilities of these statistical parameters are relatively larger as they cover different land cover types, climate zones, and surface topography. Rapid weather changes and the presence of both land and water within the grid (edge effects) may lead to the worst performance along the island/coastal stations [40]. CERES-retrieved $R_s$ data also perform relatively poorly at polar stations, which may be caused by the failure of cloud detection as more ice and snow exist in this region, and the temperature of clouds is usually not lower than that of surface snow and ice [31]. Additionally, Urraca et al. [41] pointed out the failure of most radiation products over polar regions, where strong intra-annual variations are present due to low solar elevation angles in winter, seasonal snowfall and the low viewing angle of satellites. The $R_s$ difference over the continent is observed to be smaller than that along the island/coast and polar regions, which is confirmed by the findings in other studies [42,43].

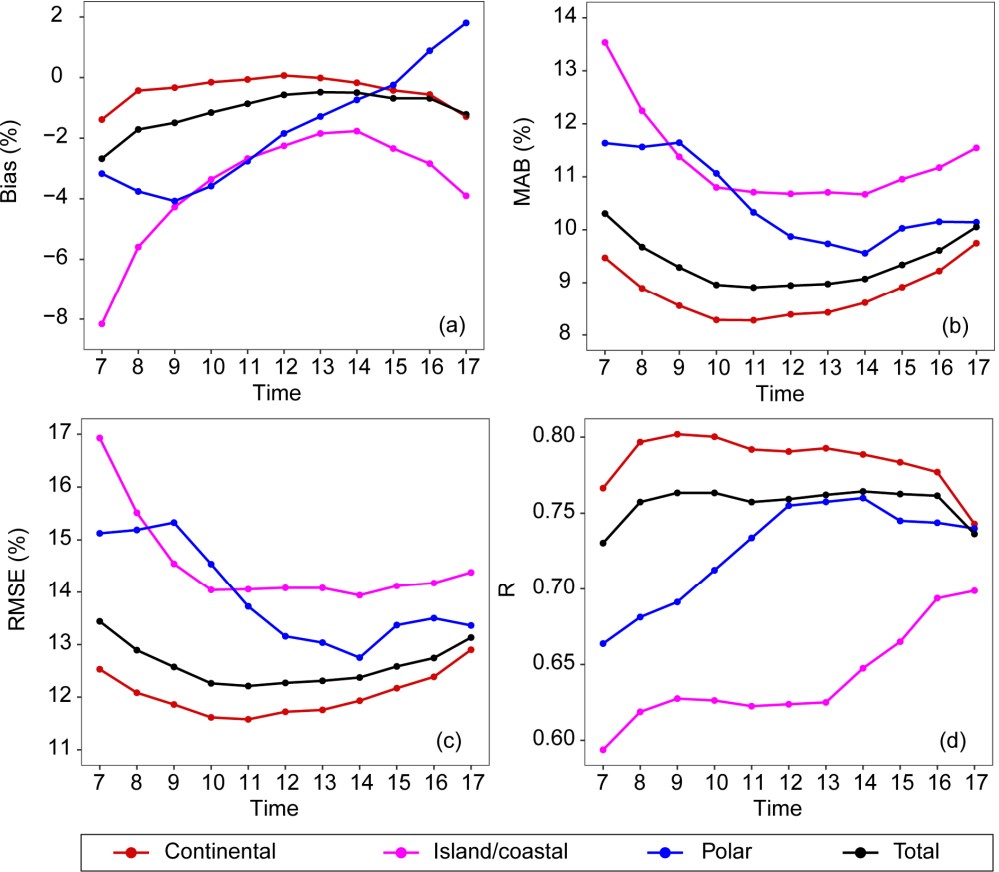

**Figure 10.** Diurnal variations of statistical parameters between hourly CERES-retrieved $R_s$ and observed $R_s$ for different types of sites. (**a**) Bias %; (**b**) Mean absolute bias (MAB) %; (**c**) Root-mean-square error (RMSE) %; and (**d**) Correlation coefficient (R).

**Table 1.** Evaluation of CERES hourly $R_s$ against the ground-based measurements for different regions. Unit: Bias %, MAB %, RMSE %.

| Region | Statistical Parameters | 7:00 | 8:00 | 9:00 | 10:00 | 11:00 | 12:00 | 13:00 | 14:00 | 15:00 | 16:00 | 17:00 |
|---|---|---|---|---|---|---|---|---|---|---|---|---|
| Continental | Bias | −1.38 | −0.43 | −0.33 | −0.15 | −0.06 | 0.07 | −0.01 | −0.17 | −0.42 | −0.56 | −1.28 |
| | MAB | 9.30 | 8.72 | 8.36 | 8.14 | 8.24 | 8.38 | 8.43 | 8.59 | 8.80 | 9.11 | 9.66 |
| | RMSE | 12.36 | 12.05 | 11.82 | 11.52 | 11.44 | 11.56 | 11.60 | 11.74 | 11.86 | 11.97 | 12.51 |
| | R | 0.77 | 0.80 | 0.80 | 0.80 | 0.78 | 0.78 | 0.79 | 0.79 | 0.79 | 0.78 | 0.74 |
| Island/coastal | Bias | −8.16 | −5.59 | −4.27 | −3.36 | −2.67 | −2.25 | −1.84 | −1.76 | −2.34 | −2.84 | −3.90 |
| | MAB | 13.54 | 12.25 | 11.38 | 10.80 | 10.71 | 10.68 | 10.71 | 10.67 | 10.96 | 11.17 | 11.55 |
| | RMSE | 16.93 | 15.51 | 14.54 | 14.04 | 14.06 | 14.08 | 14.08 | 13.94 | 14.12 | 14.18 | 14.38 |
| | R | 0.59 | 0.62 | 0.63 | 0.63 | 0.62 | 0.62 | 0.63 | 0.65 | 0.67 | 0.70 | 0.70 |
| Polar | Bias | −3.17 | −3.76 | −4.07 | −3.58 | −2.76 | −1.84 | −1.28 | −0.74 | −0.25 | 0.89 | 1.81 |
| | MAB | 12.11 | 11.92 | 11.97 | 11.27 | 10.48 | 10.01 | 9.93 | 9.84 | 10.39 | 10.54 | 10.49 |
| | RMSE | 15.69 | 15.63 | 15.74 | 14.79 | 13.90 | 13.33 | 13.27 | 13.13 | 13.84 | 14.02 | 13.83 |
| | R | 0.66 | 0.68 | 0.69 | 0.72 | 0.74 | 0.76 | 0.76 | 0.76 | 0.74 | 0.74 | 0.74 |
| Total | Bias | −2.68 | −1.71 | −1.49 | −1.15 | −0.86 | −0.57 | −0.48 | −0.50 | −0.68 | −0.69 | −1.21 |
| | MAB | 10.31 | 9.67 | 9.29 | 8.96 | 8.91 | 8.95 | 8.97 | 9.07 | 9.34 | 9.61 | 10.05 |
| | RMSE | 13.44 | 12.89 | 12.57 | 12.26 | 12.21 | 12.27 | 12.31 | 12.37 | 12.58 | 12.74 | 13.13 |
| | R | 0.73 | 0.76 | 0.76 | 0.76 | 0.76 | 0.76 | 0.76 | 0.76 | 0.76 | 0.76 | 0.74 |

### 3.2. Effect of Clouds and AOD on the Bias in CERES-Retrieved Hourly $R_s$

Cloud cover and aerosol optical depth (AOD) are two important factors that regulate $R_s$ [13,44]. Some studies have shown that CERES-retrieved cloud cover is relatively accurate [34,45–47] and AOD has high accuracy with AERONET observations [48,49]. This section focuses on the CERES-retrieved $R_s$ bias under different cloud cover and AOD (at 550 nm) categories.

Here, we define cloud cover of less than 20% as clear-sky, greater than 80% as overcast-sky and everything else as cloudy-sky. The Bias is smallest under clear-sky conditions as shown in Figure 11a, with the smallest value of approximately 0% at 9:00, slightly underestimated at midday (−0.16%) and overestimated at 14:00–17:00 (0.60%). Large underestimations in $R_s$ were found under other two conditions especially for cloudy-sky conditions, with smaller negative Bias at midday (−1.36% for cloudy-sky, −0.09% for overcast-sky) and larger negative Bias at 7:00 and 17:00 (−3.84% for cloudy-sky, −1.92% for overcast-sky).

MAB and RMSE for all times under clear-sky conditions are significantly (3.33–7.01% for MAB and 5.84–10.31% for RMSE) smaller than those under cloudy-sky (9.96–11.91% for MAB and 13.42–15.20% for RMSE) and overcast-sky conditions (10.32–11.09% for MAB and 13.66–14.67% for RMSE), especially around midday (Figure 11b,c). Compared with cloudy-sky conditions, MAB and RMSE under overcast-sky conditions are 0.26–1.10% and 0.10–0.33% larger for 9:00–15:00. The diurnal cycle of MAB and RMSE is strongest (1.30% variation and 1.59% variation) under clear-sky conditions with low values (3.33% for MAB and 5.84% for RMSE) at midday and high values (7.01% for MAB and 10.31% for RMSE) at 7:00–9:00 and 15:00–17:00. They are rather stable for cloudy-sky (0.63% variation for MAB and 0.59% variation for RMSE) and overcast-sky (0.22% variation for MAB and 0.29% variation for RMSE) conditions.

Figure 11d shows that R under overcast-sky conditions (0.75–0.77) is higher than that under clear-sky conditions (0.69–0.74) for 9:00–15:00. CERES-retrieved $R_s$ under cloudy-sky conditions has the lowest R for each hour, ranging from 0.49 at midday to ~0.60 at 7:00–9:00 and 16:00–17:00. MAB and RMSE under cloudy-sky conditions are similar to overcast-sky and all-sky conditions, but R is much lower. This may be attributed to the Bias under cloudy-sky conditions being 1.44% and 1.17% smaller than those under overcast-sky and all-sky conditions.

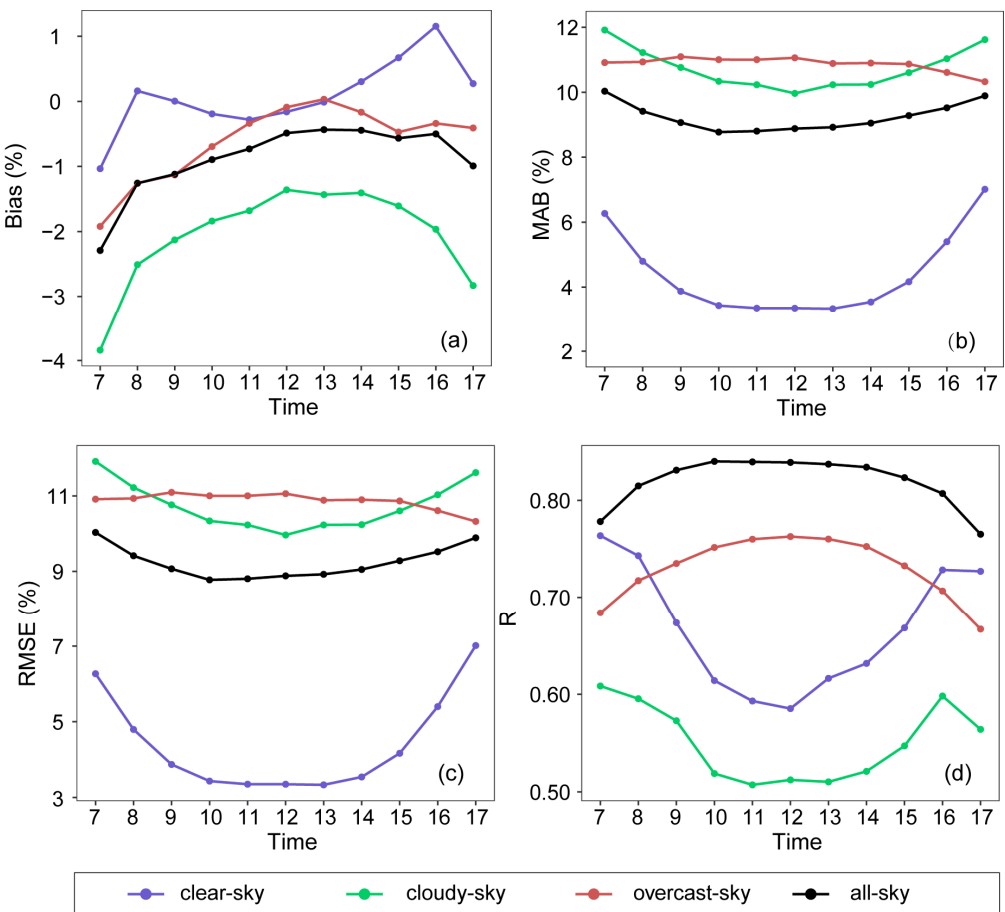

**Figure 11.** Diurnal variations of statistical parameters for CERES retrieved hourly $R_s$ at all ground-based sites for different cloud cover conditions. (**a**) Bias %; (**b**) Mean absolute bias (MAB) %; (**c**) Root-mean-square error (RMSE) %; and (**d**) Correlation coefficient (R).

R is lower at midday and higher at 7:00 and 17:00 under clear-sky and cloudy-sky conditions, while it is the opposite under overcast-sky conditions. Under clear-sky and cloudy-sky conditions, MAB and RMSE decrease/increase while R also decreases/increases correspondingly, which is the opposite of the situation presented in Figure 10 (MAB and RMSE increase while R decreases and vice versa). This may be because more clouds are likely to appear at midday [50], and there are many broken clouds in low clouds. Broken clouds are smaller in size and more disperse in the sky, and their area may be much smaller than the single pixel of CERES satellite data. Thus, many broken clouds at these hours can easily be misjudged as clear-sky or cloudy-sky conditions. It is easier for the satellite to detect the circumstances under overcast-sky conditions. R is higher for all-sky conditions with more fluctuation than that for other conditions because $R_s$ is more stable in the same cloud category.

Furthermore, we classified cloud cover more specifically into "0–20", "20–40", "40–60", "60–80", and "80–100" categories. The AOD values were divided into "0–0.05", "0.05–0.1", "0.1–0.15", "0.15–0.3" and "0.3–8" categories (see Figures 12 and 13). A negative Bias of $R_s$ were found at almost all hours. Bias was the smallest, nearly zero, in the "0–20" and "80–100" cloud cover categories. This means that the $R_s$ Bias is small under clear-sky conditions and cloudy-sky conditions. Bias, calculated by the mean method, is largest in the "60–80" cloud cover category for all hours, with the largest value of −4.71% at 7:00. Median values show that a larger negative Bias is shown in the "40–60" and "60–80" cloud cover categories, with the highest value of −5.38% at 7:00 for the "40–60" interval. At 17:00, the difference between the mean and the median values is the smallest. From

Table 2, we found changes in cloud cover contribute to 1.97–5.38% changes in $R_s$ Bias, and the variation in $R_s$ Biases caused by cloud cover is in the range of 0.86–2.43%. The change/variation in the median values (4.39–5.38%/1.98–2.43%) is larger than that in the mean values (1.97–4.03%/0.86–1.75%) as shown in Table 2, which may be due to the positive values being offset by the negative values for mean calculations.

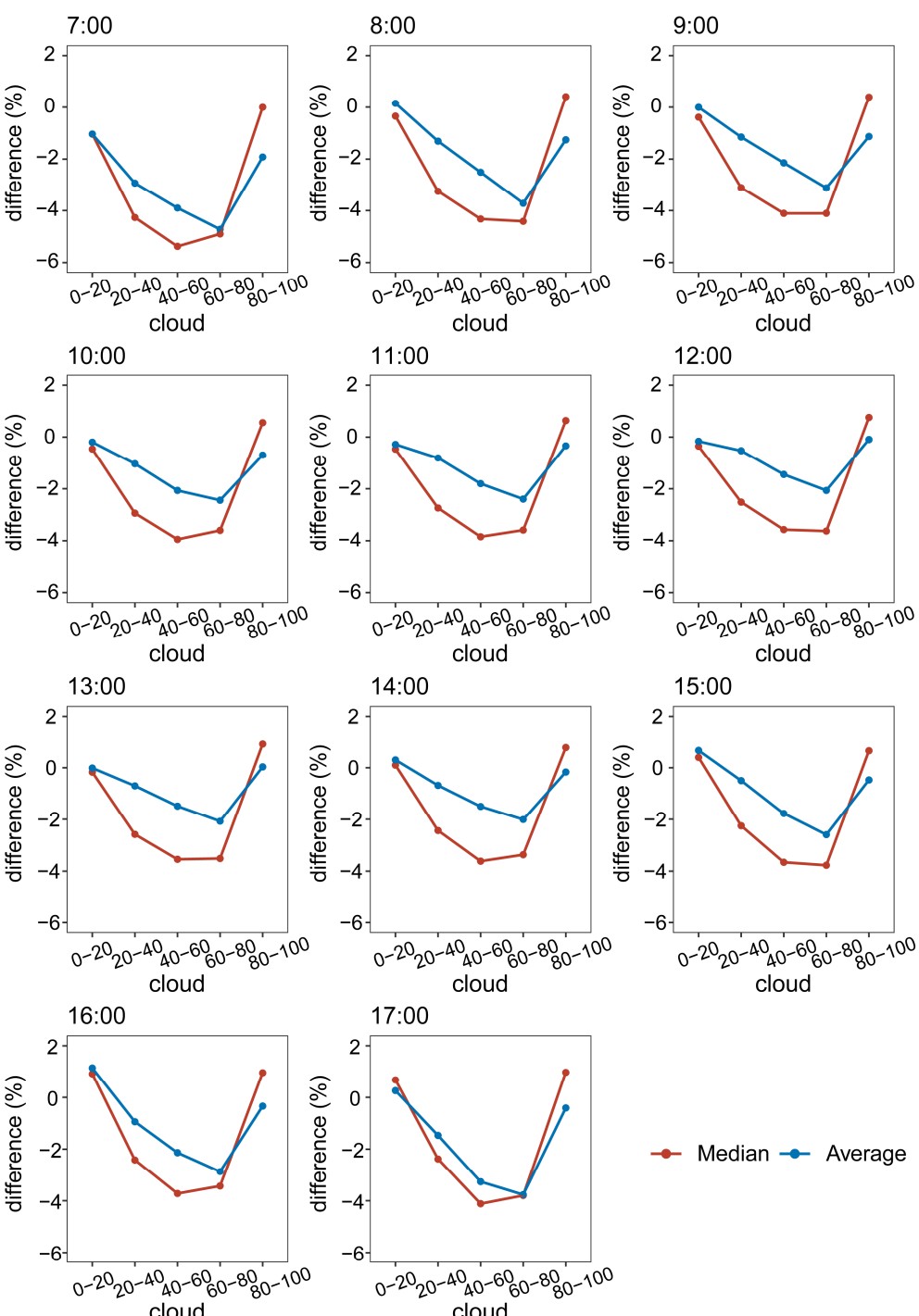

**Figure 12.** Median and mean values of differences between CERES and BSRN hourly $R_s$ under different cloud cover categories from 7:00 to 17:00. The differences refer to the values that CERES data minus BSRN data.

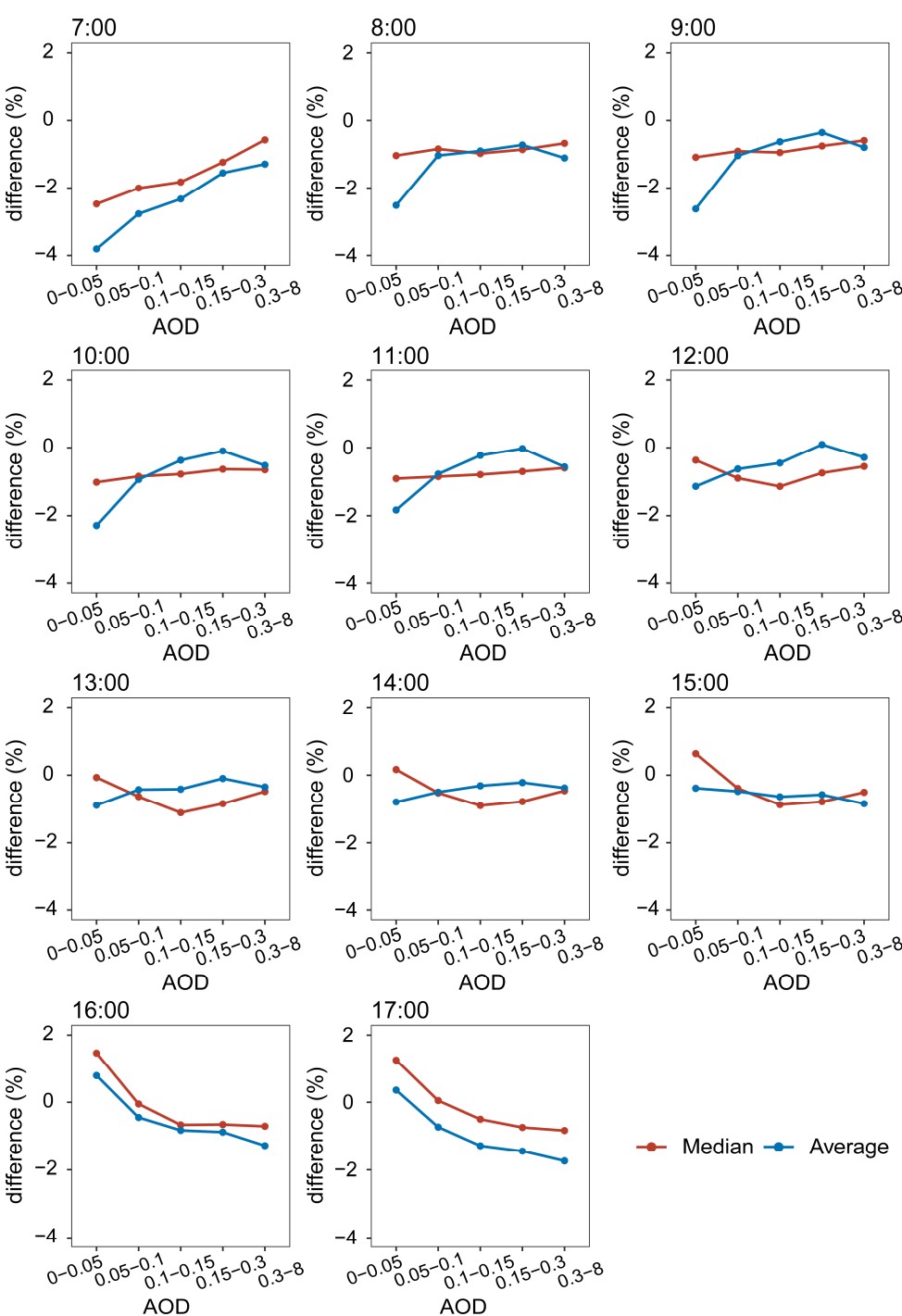

**Figure 13.** Median and mean values of differences between CERES and BSRN hourly $R_s$ under different AOD categories from 7:00 to 17:00. The differences refer to the values that CERES data minus BSRN data.

**Table 2.** Ranges (largest bias minus smallest bias) in median Biases or average Biases (shown in Figures 12 and 13) from CERES-retrieved hourly $R_s$ at all ground-based sites for each hour under different cloud cover and AOD conditions. Variation (the standard deviation of the biases) in the median Biases or average Biases shown in parentheses. Unit: %.

| Impact Factor | Range (Variation) | 7:00 | 8:00 | 9:00 | 10:00 | 11:00 | 12:00 | 13:00 | 14:00 | 15:00 | 16:00 | 17:00 |
|---|---|---|---|---|---|---|---|---|---|---|---|---|
| cloud cover | Median Biases | 5.38 | 4.83 | 4.50 | 4.51 | 4.49 | 4.39 | 4.46 | 4.39 | 4.43 | 4.68 | 5.09 |
| | | (2.43) | (2.27) | (2.13) | (2.01) | (1.99) | (1.98) | (2.04) | (2.02) | (2.14) | (2.31) | (2.42) |
| | Average Biases | 3.68 | 3.87 | 3.12 | 2.25 | 2.12 | 1.97 | 2.11 | 2.31 | 3.26 | 4.01 | 4.03 |
| | | (1.48) | (1.46) | (1.18) | (0.94) | (0.94) | (0.86) | (0.92) | (0.94) | (1.27) | (1.57) | (1.75) |
| AOD | Median Biases | 1.90 | 0.36 | 0.50 | 0.38 | 0.31 | 0.77 | 1.04 | 1.06 | 1.50 | 2.17 | 2.09 |
| | | (0.74) | (0.14) | (0.19) | (0.16) | (0.12) | (0.30) | (0.39) | (0.41) | (0.60) | (0.93) | (0.86) |
| | Average Biases | 2.52 | 1.79 | 2.27 | 2.19 | 1.80 | 1.23 | 0.79 | 0.57 | 0.46 | 2.08 | 2.10 |
| | | (1.01) | (0.72) | (0.89) | (0.86) | (0.70) | (0.45) | (0.29) | (0.22) | (0.17) | (0.80) | (0.83) |

For the mean, at 7:00, the negative Bias decreases significantly with increasing AOD; from 8:00 to 14:00, the negative Bias decreases with increasing AOD for the "0–0.3" range and then increases for the "0.3–8" range; from 16:00 to 17:00, the negative Bias increases with increasing AOD. The median is generally similar to the mean except for the "0–0.05" interval. AOD changes lead to a change of 0.31–2.52% and a variation of 0.12–1.01% in the $R_s$ Biases (Table 2). The use of median and mean values derived similar results.

In general, the change in $R_s$ Bias caused by AOD at each hour is significantly smaller than that caused by cloud cover, which in turn supports that the diurnal variation in Biases in CERES-retrieved $R_s$ is more sensitive to cloud cover than AOD.

## 4. Discussion

Based on the observed $R_s$ data at different sites in the BSRN observational network, this study evaluates CERES-retrieved hourly $R_s$ data and explores the impact of cloud cover and AOD on the bias of CERES-retrieved $R_s$ data. The impact of diurnal and seasonal variation on the $R_s$ evaluation was removed by dividing the solar radiation at TOA.

The Bias, MAB and RMSE values over the continent are 2.37%, 2.36% and 2.55% smaller than those over the island/coast and polar regions, respectively, while R is 0.1 higher. This may be attributed to rapid weather changes and the presence of both land and water within the grid (edge effects) for island/coast regions, and the failure of cloud detection as more ice and snow exist for polar regions. The spatial distribution of MAB and RMSE is consistent with the results of Yang et al. [27] and Tang et al. [22]. The Bias, MAB, RMSE at 7:00 are relatively large, then decrease at midday and increase again at 17:00. R is higher at midday than at 7:00 and 17:00. The diurnal variation in R is identical to that in Hao et al. [40], but other statistical parameters are different because we removed the diurnal cycle and seasonal cycle in the $R_s$. MAB and RMSE for all times under clear-sky conditions are significantly 3.31–7.71% and 3.35–8.59% smaller than those under other conditions, especially around midday. Bias under clear-sky conditions is about 1.47% smaller for overcast-sky conditions (except for nearly the same at 11:00–13:00) and 2.13% smaller for cloudy-sky conditions. However, R under clear-sky conditions is lower than that under overcast-sky conditions for 9:00–15:00. R under cloudy-sky conditions is significantly lower than those under other conditions. This may be because more clouds tend to occur at midday, and during these hours, many broken clouds can easily be misjudged as clear-sky or cloudy-sky conditions. Satellites are more likely to detect circumstances under overcast-sky conditions. The change in $R_s$ bias caused by AOD is 0.31–2.52% at all hours, which is significantly smaller than 1.97–5.38% caused by cloud cover. This is consistent with our previous studies as cloud cover drive the short-term variation in $R_s$ and aerosols modulate its long-term variation [51,52].

## 5. Conclusions

Satellite-retrieved datasets have relatively high accuracy and continuity in spatial distribution compared to traditional ground-based observations and reanalyses datasets. However, the evaluations of satellite-retrieved $R_s$ data on the diurnal scale are still lacking. In this study, CERES-retrieved $R_s$ on the diurnal scale was evaluated by comparison with the BSRN observations during a 21-year period from 2000 to 2021, and the influence of clouds and aerosols on the diurnal variation in $R_s$ was investigated. The findings of this study included the following:

1.  CERES-retrieved $R_s$ performs better at 11:00–13:00 ($-0.55\%$ for Bias, 8.87% for MAB, 12.58% for RMSE, and 0.84 for R) than at other hours (1.26% for Bias, 10.00% for MAB, 13.50% for RMSE, and 0.81 for R).
2.  For spatial distribution, CERES-retrieved $R_s$ performs better over the continent ($-0.42\%$ for Bias, 8.89% for MAB, 12.12% for RMSE, and 0.83 for R) than over the island/coast ($-1.01\%$ for Bias, 9.38% for MAB, 13.00% for RMSE, and 0.74 for R) and polar ($-1.70\%$ for Bias, 10.85% for MAB, 14.30% for RMSE and 0.72 for R) regions.
3.  The Bias, MAB, and RMSE in CERES-retrieved $R_s$ under clear-sky conditions are rather small, although the correlation coefficients are slightly lower than those under overcast-sky conditions from 9:00 to 15:00. R in CERES-retrieved $R_s$ under cloudy-sky conditions are the lowest.
4.  The change in $R_s$ bias caused by cloud cover is 1.97–5.38%, significantly larger than 0.31–2.52% by AOD.

In this study, the BSRN network stations used to evaluate the hourly $R_s$ data retrieved by CERES have a high accuracy. However, the observation quality of the BSRN station, the urbanization level of the city where the station is located, and whether there are natural or artificial factors that affect the observation near the station have not been further evaluated and discussed in detail. Additionally, in the BSRN observation network, there are many missing and null values at some stations, which makes the ground-based observation sequence discontinuous. Although some studies have shown that the accuracy of cloud covers and AOD in CERES is relatively high [45–49], further evaluation of their diurnal variations is needed. It is beyond the scope of this study but is essential for our future work.

**Author Contributions:** Conceptualization, Q.M.; Methodology, Q.M.; Software, L.L.; Validation, L.L. and Q.M.; Formal Analysis, Q.M.; Investigation, Q.M.; Resources, L.L.; Data Curation, L.L.; Writing—Original Draft Preparation, L.L.; Writing—Review and Editing, Q.M.; Visualization, L.L.; Supervision, Q.M. All authors have read and agreed to the published version of the manuscript.

**Funding:** This research was funded by National Science Foundation of China, grant number 41930970.

**Data Availability Statement:** Baseline Surface Radiation Network (BSRN) data are available at https://dataportals.pangaea.de/bsrn/?q=LR0100, accessed on 1 July 2021; Clouds and the Earth's Radiant Energy System for CERES SYN data are available at https://ceres.larc.nasa.gov/order_data.php, accessed on 1 July 2021.

**Acknowledgments:** We thank the following institutions for sharing their data freely: the NASA Langley Research Center Atmospheric Science Data Center for CERES SYN data; the World Data Center PANGAEA for BSRN observation data.

**Conflicts of Interest:** The authors declare no conflict of interest.

## Appendix A

**Table A1.** The information (station name, latitude (°), longitude (°) and altitude (m)) and evaluation of CERES hourly $R_s$ against the ground-based measurements for 53 stations. N is sample size of participating in the calculation.

| Site | Lat | Long | El | Statistical Parameters | 7:00 | 8:00 | 9:00 | 10:00 | 11:00 | 12:00 | 13:00 | 14:00 | 15:00 | 16:00 | 17:00 |
|---|---|---|---|---|---|---|---|---|---|---|---|---|---|---|---|
| ALE (2004.08–2014.03) | −23.8 | 133.9 | 547 | Bias | −2.04 | −2.18 | −1.69 | −0.65 | 0.1 | 1.23 | 1.96 | 2.85 | 3.12 | 4.15 | 4.82 |
| | | | | MAB | 11.12 | 11.01 | 10.93 | 10.43 | 9.85 | 9.79 | 9.82 | 9.27 | 9.40 | 9.49 | 9.40 |
| | | | | RMSE | 14.21 | 14.39 | 14.29 | 13.58 | 13.28 | 13.18 | 13.17 | 12.31 | 12.61 | 12.56 | 12.35 |
| | | | | R | 0.72 | 0.72 | 0.72 | 0.74 | 0.76 | 0.77 | 0.77 | 0.79 | 0.78 | 0.79 | 0.80 |
| | | | | N | 1638 | 1715 | 1787 | 1848 | 1886 | 1912 | 1899 | 1839 | 1779 | 1709 | 1633 |
| ASP (2000.03–2020.07) | 71.32 | −156.6 | 8 | Bias | −1.69 | −0.46 | 0.06 | 0.42 | 0.40 | 0.70 | 1.21 | 1.18 | 1.85 | 2.45 | −2.20 |
| | | | | MAB | 6.75 | 5.11 | 4.19 | 3.88 | 4.28 | 4.67 | 4.81 | 5.3 | 5.89 | 6.37 | 9.32 |
| | | | | RMSE | 9.68 | 8.14 | 7.28 | 7.02 | 7.66 | 8.06 | 8.15 | 8.74 | 9.25 | 9.47 | 12.69 |
| | | | | R | 0.78 | 0.79 | 0.82 | 0.83 | 0.82 | 0.80 | 0.84 | 0.83 | 0.82 | 0.81 | 0.70 |
| | | | | N | 6709 | 6757 | 6768 | 6778 | 6775 | 6764 | 6758 | 6756 | 6752 | 6754 | 6753 |
| BAR (2000.02–2017.08) | 32.27 | −64.67 | 8 | Bias | −3.05 | −3.10 | −3.36 | −3.37 | −2.91 | −1.91 | −0.73 | 0.79 | 2.37 | 4.29 | 5.75 |
| | | | | MAB | 11.18 | 10.87 | 10.27 | 10.12 | 9.69 | 9.36 | 9.16 | 8.96 | 8.99 | 9.45 | 9.86 |
| | | | | RMSE | 14.85 | 14.38 | 13.71 | 13.57 | 12.86 | 12.48 | 12.06 | 11.67 | 11.50 | 11.90 | 12.22 |
| | | | | R | 0.72 | 0.75 | 0.78 | 0.79 | 0.81 | 0.82 | 0.82 | 0.83 | 0.83 | 0.82 | 0.82 |
| | | | | N | 3426 | 3804 | 4126 | 4381 | 4555 | 4667 | 4615 | 4448 | 4206 | 3912 | 3570 |
| BER (2000.03–2017.08) | 36.61 | −97.52 | 317 | Bias | −1.48 | 0.33 | 0.64 | 0.48 | 0.50 | 0.24 | 0.38 | 0.33 | −0.57 | −0.43 | −1.63 |
| | | | | MAB | 8.18 | 8.25 | 8.59 | 8.57 | 8.95 | 8.94 | 9.07 | 8.78 | 8.95 | 8.69 | 9.13 |
| | | | | RMSE | 10.90 | 11.00 | 11.59 | 11.60 | 12.09 | 12.12 | 12.28 | 11.79 | 11.92 | 11.54 | 12.07 |
| | | | | R | 0.81 | 0.81 | 0.78 | 0.77 | 0.76 | 0.76 | 0.76 | 0.77 | 0.77 | 0.78 | 0.73 |
| | | | | N | 4706 | 4704 | 4697 | 4704 | 4697 | 4695 | 4697 | 4700 | 4702 | 4704 | 3994 |
| BIL (2000.03–2019.07) | 40.07 | −88.37 | 213 | Bias | −0.26 | 1.88 | 1.73 | 1.55 | 1.29 | 1.35 | 1.25 | 1.39 | 1.77 | 2.69 | 0.45 |
| | | | | MAB | 7.75 | 6.30 | 5.75 | 5.58 | 5.46 | 5.63 | 5.46 | 5.66 | 5.92 | 6.84 | 7.76 |
| | | | | RMSE | 11.03 | 8.85 | 8.41 | 8.35 | 8.09 | 8.41 | 8.00 | 8.22 | 8.44 | 9.45 | 10.83 |
| | | | | R | 0.83 | 0.90 | 0.89 | 0.87 | 0.87 | 0.87 | 0.87 | 0.86 | 0.87 | 0.87 | 0.80 |
| | | | | N | 6013 | 6775 | 6777 | 6779 | 6781 | 6779 | 6777 | 6780 | 6779 | 6779 | 5774 |
| BON (2009.01–2020.04) | 40.13 | −105.2 | 1689 | Bias | 0.49 | 0.50 | −0.94 | −1.23 | −1.46 | −1.09 | −0.81 | −0.81 | −0.62 | 0.81 | 2.09 |
| | | | | MAB | 8.24 | 8.81 | 8.54 | 8.36 | 8.41 | 8.61 | 8.51 | 8.69 | 8.47 | 8.48 | 8.80 |
| | | | | RMSE | 11.13 | 12.12 | 12.32 | 12.18 | 12.13 | 12.22 | 12.13 | 12.22 | 12.02 | 11.68 | 11.49 |
| | | | | R | 0.83 | 0.85 | 0.86 | 0.87 | 0.86 | 0.86 | 0.86 | 0.85 | 0.85 | 0.86 | 0.86 |
| | | | | N | 5183 | 7231 | 7240 | 7255 | 7269 | 7270 | 7270 | 7273 | 7273 | 7270 | 7164 |
| BOS (2009.01–2020.04) | 40.05 | −105 | 1577 | Bias | 0.46 | −0.47 | −1.18 | −1.48 | −1.41 | −1.11 | −0.93 | −0.47 | −0.45 | 0.98 | 2.63 |
| | | | | MAB | 10.98 | 10.60 | 10.48 | 10.57 | 11.53 | 11.68 | 12.31 | 12.46 | 12.27 | 11.91 | 11.44 |
| | | | | RMSE | 15.33 | 15.03 | 14.89 | 14.67 | 15.69 | 15.71 | 16.32 | 16.37 | 15.95 | 15.25 | 14.55 |
| | | | | R | 0.73 | 0.72 | 0.69 | 0.67 | 0.64 | 0.68 | 0.69 | 0.69 | 0.69 | 0.69 | 0.68 |
| | | | | N | 7280 | 7315 | 7316 | 7329 | 7332 | 7333 | 7328 | 7326 | 7327 | 7347 | 5107 |
| BOU (2000.02–2016.06) | −15.6 | −47.71 | 1023 | Bias | −1.77 | −0.72 | −1.28 | −2.09 | −2.75 | −2.76 | −2.68 | −1.28 | 0.18 | 0.95 | 3.30 |
| | | | | MAB | 12.31 | 10.88 | 10.29 | 10.28 | 11.23 | 11.59 | 12.29 | 12.30 | 12.28 | 12.14 | 11.50 |
| | | | | RMSE | 16.87 | 15.26 | 14.56 | 14.25 | 15.24 | 15.63 | 16.31 | 16.05 | 15.73 | 15.47 | 14.56 |
| | | | | R | 0.64 | 0.69 | 0.67 | 0.64 | 0.61 | 0.63 | 0.64 | 0.66 | 0.67 | 0.65 | 0.66 |
| | | | | N | 5767 | 5805 | 5809 | 5830 | 5826 | 5842 | 5847 | 5846 | 5846 | 5849 | 4083 |
| BRB (2006.02–2019.04) | 51.97 | 4.93 | 0 | Bias | −1.58 | −1.00 | −0.07 | 0.94 | 2.30 | 3.08 | 2.89 | 1.69 | 1.31 | 1.47 | 0.41 |
| | | | | MAB | 7.40 | 6.45 | 6.15 | 6.93 | 8.33 | 9.45 | 10.39 | 10.45 | 10.70 | 10.67 | 10.87 |
| | | | | RMSE | 9.82 | 8.92 | 8.99 | 10.06 | 11.88 | 13.18 | 14.18 | 14.17 | 14.29 | 14.24 | 13.90 |
| | | | | R | 0.80 | 0.81 | 0.80 | 0.75 | 0.71 | 0.69 | 0.69 | 0.70 | 0.72 | 0.72 | 0.63 |
| | | | | N | 3085 | 3086 | 3091 | 3089 | 3086 | 3083 | 3086 | 3086 | 3082 | 3084 | 3082 |
| CAB (2005.02–2021.04) | 50.22 | −5.32 | 88 | Bias | −2.56 | −0.75 | 1.30 | 0.89 | 0.92 | 1.43 | 0.93 | 1.04 | 0.64 | 0.14 | −1.47 |
| | | | | MAB | 9.62 | 9.18 | 8.76 | 8.43 | 8.40 | 8.32 | 8.45 | 8.51 | 8.60 | 8.92 | 9.15 |
| | | | | RMSE | 12.35 | 11.88 | 11.44 | 11.29 | 11.21 | 11.15 | 11.31 | 11.31 | 11.31 | 11.58 | 11.97 |
| | | | | R | 0.80 | 0.84 | 0.86 | 0.87 | 0.87 | 0.87 | 0.86 | 0.86 | 0.86 | 0.85 | 0.82 |
| | | | | N | 3939 | 5045 | 5864 | 5869 | 5866 | 5871 | 5870 | 5869 | 5871 | 5871 | 4323 |
| CAM (2001.01–2017.07) | 44.08 | 5.06 | 100 | Bias | −0.16 | −0.03 | 1.51 | 1.57 | 1.71 | 1.96 | 1.52 | 1.15 | 0.29 | −0.88 | −1.91 |
| | | | | MAB | 9.36 | 9.45 | 9.32 | 9.62 | 9.82 | 9.91 | 9.61 | 9.48 | 9.15 | 9.46 | 9.61 |
| | | | | RMSE | 12.06 | 12.19 | 12.16 | 12.57 | 12.70 | 12.94 | 12.66 | 12.46 | 11.92 | 12.19 | 12.45 |
| | | | | R | 0.80 | 0.81 | 0.83 | 0.83 | 0.83 | 0.83 | 0.83 | 0.84 | 0.85 | 0.83 | 0.81 |
| | | | | N | 4241 | 5696 | 5695 | 5696 | 5698 | 5697 | 5697 | 5697 | 5697 | 5063 | 3933 |

**Table A1.** *Cont.*

| Site | Lat | Long | El | Statistical Parameters | 7:00 | 8:00 | 9:00 | 10:00 | 11:00 | 12:00 | 13:00 | 14:00 | 15:00 | 16:00 | 17:00 |
|------|-----|------|-----|-----------------------|------|------|------|-------|-------|-------|-------|-------|-------|-------|-------|
| CAR (2000.03–2018.12) | 36.91 | −75.71 | 37 | Bias | −0.69 | −0.60 | −1.02 | −1.53 | −2.66 | −2.76 | −4.05 | −4.70 | −5.25 | −6.22 | −7.12 |
| | | | | MAB | 7.72 | 8.55 | 8.84 | 8.89 | 9.05 | 9.11 | 9.87 | 10.30 | 10.94 | 11.29 | 11.75 |
| | | | | RMSE | 10.73 | 11.90 | 12.61 | 12.81 | 12.82 | 12.89 | 13.62 | 14.19 | 15.07 | 15.22 | 15.71 |
| | | | | R | 0.83 | 0.83 | 0.79 | 0.76 | 0.74 | 0.73 | 0.72 | 0.72 | 0.72 | 0.76 | 0.72 |
| | | | | N | 4969 | 6664 | 6664 | 6663 | 6663 | 6662 | 6659 | 6658 | 6658 | 6659 | 5296 |
| CLN (2000.05–2016.11) | 42.82 | −1.6 | 471 | Bias | −5.33 | −3.04 | −2.14 | −1.48 | −1.84 | −1.93 | −2.14 | −3.13 | −4.91 | −7.60 | −9.58 |
| | | | | MAB | 9.50 | 7.94 | 7.10 | 6.76 | 6.83 | 7.27 | 7.31 | 8.53 | 10.17 | 12.56 | 14.20 |
| | | | | RMSE | 12.11 | 10.40 | 9.67 | 9.51 | 9.58 | 10.15 | 10.19 | 11.44 | 13.12 | 15.72 | 17.79 |
| | | | | R | 0.86 | 0.88 | 0.88 | 0.88 | 0.87 | 0.87 | 0.87 | 0.85 | 0.82 | 0.78 | 0.66 |
| | | | | N | 5647 | 5635 | 5619 | 5616 | 5609 | 5597 | 5605 | 5602 | 5628 | 5647 | 4069 |
| CNR (2009.07–2021.05) | −12.19 | 96.84 | 6 | Bias | 2.57 | 4.64 | 4.72 | 4.48 | 4.21 | 4.29 | 2.88 | 3.13 | 2.51 | −0.26 | −1.59 |
| | | | | MAB | 9.28 | 9.62 | 9.16 | 9.21 | 8.84 | 8.80 | 8.58 | 8.75 | 8.96 | 9.16 | 9.25 |
| | | | | RMSE | 12.47 | 13.38 | 12.85 | 13.04 | 12.61 | 12.55 | 12.02 | 12.23 | 12.18 | 12.40 | 12.29 |
| | | | | R | 0.83 | 0.86 | 0.87 | 0.86 | 0.86 | 0.86 | 0.86 | 0.86 | 0.86 | 0.83 | 0.80 |
| | | | | N | 4083 | 4328 | 4320 | 4317 | 4317 | 4322 | 4324 | 4325 | 4332 | 4331 | 2978 |
| COC (2004.10–2020.05) | −30.67 | 23.99 | 1287 | Bias | −3.20 | −0.91 | −0.21 | 0.21 | 0.24 | 0.89 | 1.05 | 1.18 | 1.19 | 0.86 | −0.21 |
| | | | | MAB | 10.13 | 9.27 | 8.70 | 8.45 | 8.64 | 8.45 | 8.52 | 8.64 | 8.66 | 8.74 | 8.39 |
| | | | | RMSE | 13.15 | 12.04 | 11.63 | 11.55 | 11.86 | 11.70 | 11.71 | 11.72 | 11.56 | 11.58 | 10.84 |
| | | | | R | 0.61 | 0.67 | 0.67 | 0.67 | 0.65 | 0.68 | 0.68 | 0.68 | 0.70 | 0.71 | 0.72 |
| | | | | N | 4209 | 4214 | 4217 | 4209 | 4197 | 4190 | 4186 | 4179 | 4172 | 4162 | 4146 |
| DAA (2000.07–2020.01) | −12.43 | 130.9 | 30 | Bias | −3.47 | 0.37 | 0.69 | 0.94 | 1.43 | 2.37 | 2.70 | 1.88 | 1.87 | 1.76 | −1.11 |
| | | | | MAB | 9.60 | 5.95 | 4.90 | 4.58 | 4.61 | 4.96 | 5.37 | 5.74 | 6.53 | 7.73 | 9.24 |
| | | | | RMSE | 13.53 | 9.93 | 8.64 | 8.42 | 8.76 | 9.50 | 9.91 | 9.72 | 10.23 | 11.26 | 12.60 |
| | | | | R | 0.75 | 0.81 | 0.80 | 0.79 | 0.78 | 0.78 | 0.80 | 0.79 | 0.76 | 0.75 | 0.77 |
| | | | | N | 3408 | 3478 | 3503 | 3509 | 3507 | 3509 | 3513 | 3516 | 3519 | 3516 | 3506 |
| DAR (2002.07–2015.01) | −75.1 | 123.4 | 3233 | Bias | −4.08 | −2.22 | −1.00 | 1.32 | 1.75 | 1.25 | 1.21 | 1.47 | −0.44 | −0.37 | −0.99 |
| | | | | MAB | 8.35 | 7.73 | 7.80 | 8.22 | 9.31 | 9.20 | 8.09 | 7.97 | 7.69 | 7.51 | 7.90 |
| | | | | RMSE | 10.93 | 10.47 | 10.71 | 11.91 | 13.10 | 12.92 | 11.99 | 11.82 | 10.99 | 10.75 | 10.56 |
| | | | | R | 0.64 | 0.67 | 0.69 | 0.67 | 0.61 | 0.64 | 0.67 | 0.66 | 0.69 | 0.69 | 0.58 |
| | | | | N | 4532 | 4533 | 4533 | 4534 | 4533 | 4533 | 4513 | 4516 | 4517 | 4517 | 4518 |
| DOM (2006.01–2021.02) | 36.63 | −116 | 1007 | Bias | −8.62 | −11.24 | −12.71 | −10.59 | −6.39 | −5.17 | −4.39 | −4.74 | −7.75 | −6.87 | −3.86 |
| | | | | MAB | 12.50 | 12.54 | 13.60 | 11.68 | 7.79 | 6.73 | 6.22 | 6.76 | 10.12 | 10.43 | 9.27 |
| | | | | RMSE | 16.68 | 16.98 | 18.21 | 15.39 | 10.68 | 9.66 | 9.25 | 9.95 | 14.45 | 15.48 | 13.95 |
| | | | | R | 0.47 | 0.48 | 0.48 | 0.53 | 0.63 | 0.69 | 0.72 | 0.70 | 0.65 | 0.68 | 0.73 |
| | | | | N | 2537 | 2752 | 2963 | 3101 | 3271 | 3394 | 3439 | 3378 | 3263 | 3116 | 2919 |
| DRA (2009.01–2020.04) | −12.42 | 130.9 | 32 | Bias | 0.89 | 1.29 | 0.58 | −0.56 | −0.77 | −0.48 | −0.51 | −0.31 | 0.63 | 2.27 | 5.03 |
| | | | | MAB | 6.96 | 6.02 | 5.24 | 4.87 | 4.96 | 5.41 | 5.42 | 5.73 | 5.82 | 6.70 | 8.89 |
| | | | | RMSE | 10.61 | 9.61 | 8.92 | 8.31 | 8.21 | 9.04 | 8.90 | 9.29 | 9.30 | 9.76 | 12.42 |
| | | | | R | 0.82 | 0.87 | 0.85 | 0.83 | 0.80 | 0.78 | 0.79 | 0.79 | 0.81 | 0.82 | 0.86 |
| | | | | N | 5692 | 7215 | 7210 | 7213 | 7214 | 7216 | 7221 | 7223 | 7223 | 7240 | 7016 |
| DWN (2008.04–2020.07) | 36.61 | −97.49 | 318 | Bias | −5.32 | −2.63 | −0.38 | 1.93 | 2.63 | 2.05 | 2.20 | 2.28 | −0.52 | −0.86 | −2.53 |
| | | | | MAB | 9.15 | 7.75 | 7.69 | 8.13 | 9.02 | 8.89 | 7.77 | 7.85 | 7.53 | 7.56 | 8.31 |
| | | | | RMSE | 11.80 | 10.17 | 10.66 | 11.93 | 13.16 | 12.77 | 11.78 | 11.81 | 10.64 | 10.69 | 10.92 |
| | | | | R | 0.64 | 0.67 | 0.69 | 0.67 | 0.59 | 0.61 | 0.66 | 0.66 | 0.68 | 0.70 | 0.60 |
| | | | | N | 3999 | 4024 | 4025 | 4030 | 4037 | 4042 | 4041 | 4039 | 4035 | 4027 | 4026 |
| E13 (2000.02–2019.07) | −27.6 | −48.52 | 11 | Bias | −0.59 | 1.58 | 1.42 | 1.12 | 0.76 | 0.84 | 0.72 | 0.86 | 1.20 | 1.99 | −0.52 |
| | | | | MAB | 7.77 | 6.26 | 5.70 | 5.54 | 5.39 | 5.58 | 5.37 | 5.56 | 5.79 | 6.61 | 7.74 |
| | | | | RMSE | 10.99 | 8.76 | 8.34 | 8.30 | 8.01 | 8.36 | 7.89 | 8.14 | 8.30 | 9.26 | 10.89 |
| | | | | R | 0.83 | 0.90 | 0.89 | 0.87 | 0.87 | 0.87 | 0.87 | 0.87 | 0.87 | 0.87 | 0.80 |
| | | | | N | 6125 | 6884 | 6894 | 6890 | 6892 | 6893 | 6894 | 6893 | 6892 | 6892 | 5987 |
| FLO (2000.04–2021.03) | 48.32 | −105.1 | 634 | Bias | 0.13 | 0.00 | 0.30 | −0.14 | 0.51 | 0.19 | −0.45 | −1.38 | −0.59 | −0.57 | 0.25 |
| | | | | MAB | 8.91 | 8.72 | 8.56 | 8.42 | 8.27 | 8.10 | 8.45 | 8.86 | 9.02 | 9.21 | 8.88 |
| | | | | RMSE | 11.67 | 11.62 | 11.69 | 11.45 | 11.38 | 10.93 | 11.32 | 11.75 | 12.24 | 12.37 | 12.08 |
| | | | | R | 0.81 | 0.85 | 0.86 | 0.86 | 0.86 | 0.87 | 0.86 | 0.86 | 0.85 | 0.83 | 0.72 |
| | | | | N | 4485 | 4501 | 4509 | 4510 | 4509 | 4508 | 4510 | 4509 | 4495 | 4476 | 4470 |
| FPE (2009.01–2020.05) | 33.58 | 130.4 | 3 | Bias | 1.26 | 1.21 | 0.06 | 0.56 | 0.69 | 0.48 | 0.00 | −0.79 | 0.48 | 2.31 | 3.33 |
| | | | | MAB | 9.00 | 8.94 | 8.73 | 8.02 | 8.02 | 8.27 | 8.60 | 9.08 | 9.41 | 9.77 | 9.90 |
| | | | | RMSE | 12.50 | 12.49 | 12.46 | 11.47 | 11.35 | 11.66 | 11.99 | 12.61 | 12.76 | 13.07 | 13.05 |
| | | | | R | 0.83 | 0.84 | 0.82 | 0.83 | 0.82 | 0.81 | 0.80 | 0.80 | 0.80 | 0.79 | 0.76 |
| | | | | N | 5631 | 6701 | 6735 | 6737 | 6747 | 6758 | 6778 | 6775 | 6746 | 5694 | 4363 |

**Table A1.** *Cont.*

| Site | Lat | Long | El | Statistical Parameters | 7:00 | 8:00 | 9:00 | 10:00 | 11:00 | 12:00 | 13:00 | 14:00 | 15:00 | 16:00 | 17:00 |
|---|---|---|---|---|---|---|---|---|---|---|---|---|---|---|---|
| FUA (2010.04–2021.04) | 34.25 | −89.87 | 98 | Bias | −0.76 | 0.71 | 0.62 | 0.55 | 0.61 | 0.75 | 0.80 | 0.40 | −0.01 | 0.21 | −2.24 |
| | | | | MAB | 7.40 | 7.22 | 7.20 | 7.38 | 7.66 | 7.88 | 7.71 | 7.55 | 7.48 | 7.59 | 8.10 |
| | | | | RMSE | 9.61 | 9.58 | 9.74 | 10.03 | 10.36 | 10.55 | 10.44 | 10.21 | 9.96 | 10.06 | 10.67 |
| | | | | R | 0.88 | 0.91 | 0.91 | 0.90 | 0.90 | 0.90 | 0.90 | 0.90 | 0.90 | 0.89 | 0.84 |
| | | | | N | 4014 | 4013 | 4013 | 4012 | 4010 | 4007 | 4010 | 4010 | 4012 | 4014 | 3232 |
| GCR (2009.01–2020.04) | −23.56 | 15.04 | 407 | Bias | 3.19 | 3.02 | 1.92 | 1.09 | 0.73 | 0.63 | 0.51 | 0.08 | 0.02 | 0.68 | 1.21 |
| | | | | MAB | 7.35 | 7.44 | 6.54 | 5.94 | 6.13 | 6.45 | 6.63 | 6.95 | 7.17 | 7.54 | 8.40 |
| | | | | RMSE | 9.58 | 9.92 | 8.92 | 8.37 | 8.71 | 9.08 | 9.45 | 9.85 | 10.08 | 10.42 | 11.35 |
| | | | | R | 0.86 | 0.89 | 0.89 | 0.90 | 0.90 | 0.89 | 0.89 | 0.88 | 0.85 | 0.85 | 0.83 |
| | | | | N | 5285 | 7093 | 7158 | 7203 | 7203 | 7204 | 7206 | 7205 | 7205 | 7193 | 7150 |
| GOB (2012.05–2021.04) | −70.65 | −8.25 | 42 | Bias | 2.00 | 4.00 | 1.26 | 1.09 | 0.75 | 0.35 | 0.62 | −0.29 | −1.54 | −4.38 | −7.75 |
| | | | | MAB | 13.91 | 10.98 | 7.23 | 4.31 | 2.94 | 2.59 | 2.49 | 2.76 | 3.54 | 6.06 | 9.20 |
| | | | | RMSE | 17.39 | 15.50 | 11.52 | 7.94 | 5.76 | 5.01 | 4.92 | 5.36 | 5.91 | 8.66 | 11.82 |
| | | | | R | 0.33 | 0.53 | 0.70 | 0.78 | 0.78 | 0.73 | 0.76 | 0.78 | 0.70 | 0.62 | 0.66 |
| | | | | N | 2903 | 3211 | 3213 | 3214 | 3214 | 3213 | 3213 | 3213 | 3215 | 3215 | 3214 |
| GVN (2000.03–2021.01) | 24.34 | 124.2 | 5.7 | Bias | −13.25 | −13.34 | −13.45 | −12.15 | −10.83 | −8.67 | −8.06 | −6.71 | −4.80 | −2.66 | −1.34 |
| | | | | MAB | 16.68 | 16.30 | 16.19 | 15.01 | 14.06 | 12.60 | 12.24 | 11.54 | 10.94 | 10.45 | 10.06 |
| | | | | RMSE | 20.97 | 20.40 | 20.02 | 18.69 | 17.87 | 16.05 | 15.84 | 14.85 | 14.17 | 13.64 | 13.21 |
| | | | | R | 0.54 | 0.59 | 0.62 | 0.65 | 0.67 | 0.70 | 0.70 | 0.72 | 0.71 | 0.70 | 0.70 |
| | | | | N | 3831 | 4381 | 4963 | 5365 | 5736 | 5925 | 5942 | 5640 | 5298 | 4860 | 4364 |
| ISH (2010.04–2021.04) | 28.31 | −16.5 | 2372.9 | Bias | −1.25 | −0.85 | −0.55 | −0.27 | 0.44 | 0.86 | 1.16 | 0.94 | −0.23 | −1.18 | −2.66 |
| | | | | MAB | 7.12 | 7.45 | 7.57 | 8.11 | 8.09 | 8.56 | 8.72 | 8.67 | 8.65 | 8.44 | 8.66 |
| | | | | RMSE | 9.51 | 10.03 | 10.27 | 11.10 | 11.19 | 11.92 | 12.15 | 12.05 | 11.74 | 11.35 | 11.56 |
| | | | | R | 0.85 | 0.87 | 0.87 | 0.85 | 0.84 | 0.83 | 0.83 | 0.83 | 0.84 | 0.85 | 0.84 |
| | | | | N | 4009 | 4007 | 4006 | 4003 | 4006 | 4006 | 4004 | 4005 | 4006 | 4008 | 4008 |
| IZA (2009.03–2021.06) | 8.72 | 167.7 | 10 | Bias | −25.76 | −22.14 | −19.26 | −16.82 | −15.28 | −14.21 | −14.72 | −14.97 | −17.67 | −22.17 | −27.90 |
| | | | | MAB | 26.60 | 23.36 | 20.91 | 19.26 | 18.60 | 18.26 | 18.57 | 18.81 | 21.04 | 24.55 | 28.93 |
| | | | | RMSE | 28.98 | 25.83 | 23.43 | 21.68 | 20.80 | 20.51 | 20.85 | 21.01 | 23.24 | 26.93 | 31.66 |
| | | | | R | 0.48 | 0.24 | 0.19 | 0.19 | 0.22 | 0.23 | 0.26 | 0.34 | 0.39 | 0.53 | 0.52 |
| | | | | N | 4441 | 4435 | 4439 | 4442 | 4445 | 4444 | 4443 | 4441 | 4444 | 4447 | 3662 |
| KWA (2000.03–2017.08) | −45.05 | 169.7 | 350 | Bias | −3.74 | −0.82 | −0.49 | −0.31 | −0.34 | −0.95 | −0.24 | −0.02 | 0.91 | 0.77 | 1.11 |
| | | | | MAB | 10.12 | 9.11 | 8.62 | 8.13 | 8.06 | 8.56 | 8.13 | 8.12 | 7.97 | 7.76 | 7.75 |
| | | | | RMSE | 13.09 | 12.08 | 11.60 | 11.19 | 11.14 | 11.78 | 11.17 | 11.20 | 10.91 | 10.44 | 10.22 |
| | | | | R | 0.55 | 0.67 | 0.71 | 0.72 | 0.71 | 0.71 | 0.72 | 0.74 | 0.74 | 0.75 | 0.75 |
| | | | | N | 5945 | 5934 | 5872 | 5669 | 5585 | 5480 | 5596 | 5645 | 5904 | 5958 | 5965 |
| LAU (2000.03–2018.12) | 60.14 | −1.18 | 80 | Bias | −7.58 | −7.80 | −5.14 | −4.02 | −3.64 | −3.58 | −3.14 | −3.10 | −3.35 | −1.59 | −1.00 |
| | | | | MAB | 15.93 | 16.05 | 14.44 | 13.59 | 13.66 | 13.28 | 13.05 | 13.10 | 13.22 | 12.14 | 11.38 |
| | | | | RMSE | 20.59 | 21.00 | 18.30 | 17.25 | 17.53 | 17.24 | 17.06 | 17.18 | 17.43 | 16.07 | 15.02 |
| | | | | R | 0.57 | 0.58 | 0.68 | 0.69 | 0.67 | 0.68 | 0.68 | 0.68 | 0.67 | 0.69 | 0.70 |
| | | | | N | 4319 | 6132 | 6319 | 6364 | 6391 | 6413 | 6411 | 6409 | 6411 | 6420 | 5318 |
| LER (2001.01–2017.07) | 52.21 | 14.12 | 125 | Bias | 0.12 | −0.63 | −1.74 | −1.01 | −1.18 | −0.48 | −0.27 | −0.54 | 0.58 | 1.42 | 2.20 |
| | | | | MAB | 11.42 | 11.30 | 11.46 | 10.85 | 10.73 | 10.71 | 10.80 | 11.50 | 11.69 | 11.67 | 11.60 |
| | | | | RMSE | 14.13 | 14.24 | 14.86 | 13.92 | 13.87 | 13.73 | 13.80 | 14.69 | 14.83 | 14.66 | 14.34 |
| | | | | R | 0.71 | 0.74 | 0.73 | 0.78 | 0.78 | 0.79 | 0.78 | 0.73 | 0.72 | 0.69 | 0.66 |
| | | | | N | 3748 | 4378 | 5375 | 5378 | 5380 | 5379 | 5381 | 5381 | 4484 | 3872 | 3235 |
| LIN (2000.03–2018.12) | −2.06 | 147.4 | 6 | Bias | −1.67 | −1.50 | −1.11 | −1.96 | −1.77 | −1.67 | −2.20 | −2.38 | −3.57 | −5.05 | −6.92 |
| | | | | MAB | 8.81 | 9.19 | 9.22 | 9.09 | 9.23 | 9.35 | 9.56 | 9.58 | 10.69 | 11.22 | 12.20 |
| | | | | RMSE | 11.24 | 11.93 | 12.09 | 12.10 | 12.08 | 12.43 | 12.67 | 12.73 | 14.04 | 14.67 | 15.58 |
| | | | | R | 0.86 | 0.85 | 0.86 | 0.86 | 0.86 | 0.85 | 0.84 | 0.83 | 0.79 | 0.75 | 0.70 |
| | | | | N | 3035 | 3893 | 3892 | 3894 | 3893 | 3893 | 3893 | 3894 | 3894 | 3031 | 2418 |
| LRC (2014.12–2021.05) | 24.29 | 154 | 7.1 | Bias | −3.87 | −2.66 | −1.69 | −1.04 | −0.27 | 0.69 | 1.75 | 1.99 | 1.58 | −0.50 | −2.30 |
| | | | | MAB | 10.37 | 10.23 | 10.20 | 11.03 | 11.68 | 12.39 | 12.56 | 12.24 | 11.67 | 10.83 | 9.78 |
| | | | | RMSE | 13.23 | 13.20 | 13.41 | 14.57 | 15.36 | 16.17 | 16.60 | 16.18 | 15.21 | 13.95 | 12.56 |
| | | | | R | 0.70 | 0.73 | 0.73 | 0.71 | 0.71 | 0.70 | 0.68 | 0.69 | 0.70 | 0.69 | 0.60 |
| | | | | N | 4887 | 4891 | 4896 | 4889 | 4890 | 4889 | 4889 | 4890 | 4888 | 4886 | 4886 |
| MAN (2000.03–2013.10) | −0.52 | 166.9 | 7 | Bias | −8.90 | −3.62 | −1.82 | −0.69 | 0.04 | 0.36 | 0.37 | 0.63 | 0.23 | −0.71 | −0.93 |
| | | | | MAB | 12.74 | 9.55 | 8.39 | 7.47 | 7.16 | 7.32 | 7.02 | 7.20 | 7.28 | 7.36 | 7.07 |
| | | | | RMSE | 16.46 | 12.61 | 11.50 | 10.67 | 10.63 | 10.91 | 10.46 | 10.54 | 10.32 | 10.10 | 9.37 |
| | | | | R | 0.58 | 0.70 | 0.70 | 0.70 | 0.71 | 0.69 | 0.71 | 0.72 | 0.74 | 0.76 | 0.82 |
| | | | | N | 3975 | 3976 | 3977 | 3975 | 3969 | 3964 | 3958 | 3954 | 3959 | 3966 | 3972 |

**Table A1.** *Cont.*

| Site | Lat | Long | El | Statistical Parameters | 7:00 | 8:00 | 9:00 | 10:00 | 11:00 | 12:00 | 13:00 | 14:00 | 15:00 | 16:00 | 17:00 |
|------|-----|------|-----|------------------------|------|------|------|-------|-------|-------|-------|-------|-------|-------|-------|
| MNM (2010.04–2021.04) | 78.93 | 11.93 | 11 | Bias | −4.64 | −1.01 | −0.46 | 0.72 | 1.76 | 2.44 | 3.48 | 3.96 | 4.38 | 4.23 | 2.99 |
|  |  |  |  | MAB | 9.36 | 8.13 | 7.75 | 7.90 | 8.39 | 8.67 | 9.24 | 9.50 | 9.53 | 9.36 | 8.97 |
|  |  |  |  | RMSE | 12.23 | 11.01 | 10.82 | 11.52 | 12.45 | 12.69 | 13.33 | 13.50 | 13.12 | 12.55 | 11.77 |
|  |  |  |  | R | 0.55 | 0.64 | 0.61 | 0.59 | 0.57 | 0.56 | 0.55 | 0.55 | 0.58 | 0.60 | 0.63 |
|  |  |  |  | N | 4857 | 4860 | 4861 | 4865 | 4863 | 4866 | 4869 | 4865 | 4864 | 4865 | 4866 |
| NAU (2000.03–2013.09) | 48.71 | 2.21 | 156 | Bias | −9.97 | −8.79 | −7.43 | −6.45 | −4.67 | −3.18 | −1.93 | −2.12 | −3.84 | −3.68 | −3.65 |
|  |  |  |  | MAB | 15.22 | 14.25 | 13.62 | 13.03 | 12.23 | 11.96 | 12.06 | 11.21 | 10.99 | 10.79 | 10.75 |
|  |  |  |  | RMSE | 20.03 | 18.54 | 17.48 | 16.88 | 15.94 | 15.71 | 15.79 | 14.58 | 14.45 | 14.19 | 14.09 |
|  |  |  |  | R | 0.60 | 0.64 | 0.67 | 0.68 | 0.69 | 0.68 | 0.64 | 0.70 | 0.73 | 0.73 | 0.72 |
|  |  |  |  | N | 4299 | 4601 | 4864 | 5057 | 5183 | 5267 | 5200 | 5040 | 4839 | 4579 | 4278 |
| NYA (2000.03–2021.07) | 46.82 | 6.94 | 491 | Bias | −0.65 | 0.77 | 2.35 | 1.87 | 1.21 | 1.08 | 0.66 | 0.43 | 0.47 | −0.24 | −2.34 |
|  |  |  |  | MAB | 8.23 | 8.52 | 8.74 | 8.18 | 8.64 | 8.67 | 9.04 | 8.87 | 9.16 | 9.69 | 10.09 |
|  |  |  |  | RMSE | 10.66 | 11.31 | 11.74 | 11.11 | 11.62 | 11.45 | 12.00 | 11.74 | 12.10 | 12.58 | 13.12 |
|  |  |  |  | R | 0.85 | 0.86 | 0.87 | 0.88 | 0.86 | 0.87 | 0.85 | 0.86 | 0.84 | 0.81 | 0.77 |
|  |  |  |  | N | 3363 | 4435 | 4997 | 5000 | 5003 | 5001 | 5001 | 4998 | 4996 | 4995 | 3950 |
| PAL (2005.10–2019.12) | 40.72 | −77.93 | 376 | Bias | 1.39 | 3.31 | 3.28 | 2.72 | 1.89 | 2.01 | 0.74 | 0.66 | 0.72 | 0.41 | −1.12 |
|  |  |  |  | MAB | 9.26 | 10.38 | 10.89 | 10.67 | 10.74 | 10.37 | 10.23 | 10.25 | 10.20 | 10.05 | 9.51 |
|  |  |  |  | RMSE | 12.25 | 13.81 | 14.78 | 14.71 | 14.66 | 14.23 | 13.86 | 13.84 | 13.67 | 13.13 | 12.56 |
|  |  |  |  | R | 0.83 | 0.81 | 0.80 | 0.80 | 0.79 | 0.80 | 0.80 | 0.80 | 0.81 | 0.82 | 0.81 |
|  |  |  |  | N | 5548 | 7500 | 7497 | 7497 | 7496 | 7503 | 7502 | 7505 | 7508 | 7507 | 5674 |
| PAY (2000.03–2020.12) | −9.07 | −40.32 | 387 | Bias | 4.79 | 2.61 | 2.16 | 1.59 | 1.45 | 1.26 | 0.74 | 0.53 | 0.71 | 1.23 | 0.51 |
|  |  |  |  | MAB | 9.37 | 8.17 | 8.20 | 8.13 | 8.40 | 8.62 | 8.63 | 8.57 | 8.76 | 9.10 | 9.36 |
|  |  |  |  | RMSE | 12.08 | 11.25 | 11.37 | 11.28 | 11.51 | 11.80 | 11.82 | 11.73 | 11.76 | 12.05 | 12.38 |
|  |  |  |  | R | 0.85 | 0.88 | 0.89 | 0.89 | 0.88 | 0.88 | 0.87 | 0.87 | 0.86 | 0.84 | 0.78 |
|  |  |  |  | N | 7244 | 7228 | 7223 | 7228 | 7226 | 7227 | 7232 | 7230 | 7232 | 7225 | 5216 |
| PSU (2009.01–2020.04) | 50.21 | −104.7 | 578 | Bias | 1.41 | 2.98 | 4.36 | 5.07 | 5.09 | 4.62 | 3.50 | 2.36 | 1.75 | 0.88 | 2.49 |
|  |  |  |  | MAB | 9.26 | 9.56 | 9.53 | 9.17 | 8.70 | 8.36 | 7.87 | 7.66 | 7.82 | 7.96 | 8.27 |
|  |  |  |  | RMSE | 11.96 | 12.49 | 12.37 | 11.99 | 11.40 | 10.89 | 10.24 | 10.01 | 10.31 | 10.53 | 11.32 |
|  |  |  |  | R | 0.64 | 0.69 | 0.71 | 0.73 | 0.73 | 0.73 | 0.69 | 0.66 | 0.64 | 0.63 | 0.61 |
|  |  |  |  | N | 3246 | 3275 | 3286 | 3287 | 3289 | 3287 | 3285 | 3284 | 3289 | 3288 | 3287 |
| PTR (2006.12–2018.07) | 43.06 | 141.3 | 17.2 | Bias | −6.64 | −5.18 | −4.08 | −3.52 | −2.97 | −2.30 | −2.61 | −2.87 | −3.85 | −2.54 | −1.69 |
|  |  |  |  | MAB | 12.54 | 11.56 | 10.90 | 9.94 | 9.23 | 8.98 | 9.27 | 9.78 | 10.26 | 9.44 | 9.09 |
|  |  |  |  | RMSE | 16.30 | 15.70 | 14.78 | 13.87 | 13.01 | 12.67 | 13.30 | 13.88 | 14.62 | 13.28 | 12.58 |
|  |  |  |  | R | 0.72 | 0.74 | 0.77 | 0.80 | 0.81 | 0.81 | 0.79 | 0.78 | 0.78 | 0.79 | 0.78 |
|  |  |  |  | N | 2705 | 3448 | 4277 | 4289 | 4298 | 4304 | 4307 | 4307 | 4308 | 4306 | 3471 |
| REG (2000.02–2011.12) | 30.86 | 34.78 | 500 | Bias | −5.96 | −7.76 | −7.38 | −6.64 | −5.81 | −4.25 | −3.80 | −3.64 | −3.97 | −4.46 | −4.29 |
|  |  |  |  | MAB | 11.81 | 13.33 | 13.67 | 13.54 | 13.57 | 13.14 | 13.04 | 12.92 | 12.51 | 11.70 | 10.83 |
|  |  |  |  | RMSE | 15.19 | 17.51 | 18.16 | 18.13 | 18.28 | 17.74 | 17.59 | 17.56 | 16.99 | 15.75 | 14.29 |
|  |  |  |  | R | 0.72 | 0.68 | 0.68 | 0.68 | 0.66 | 0.66 | 0.67 | 0.67 | 0.68 | 0.69 | 0.72 |
|  |  |  |  | N | 3024 | 3872 | 3873 | 3875 | 3872 | 3875 | 3875 | 3873 | 3873 | 3878 | 3089 |
| SAP (2010.04–2020.11) | −29.44 | −53.82 | 489 | Bias | 1.76 | 2.84 | 2.51 | 2.40 | 1.87 | 2.44 | 2.20 | 2.47 | 3.46 | 4.87 | 7.86 |
|  |  |  |  | MAB | 7.35 | 6.08 | 5.46 | 4.85 | 4.45 | 4.31 | 4.33 | 4.68 | 5.25 | 6.39 | 8.89 |
|  |  |  |  | RMSE | 10.07 | 8.99 | 8.61 | 8.00 | 7.45 | 7.63 | 7.47 | 7.85 | 8.28 | 8.95 | 11.80 |
|  |  |  |  | R | 0.80 | 0.83 | 0.81 | 0.81 | 0.82 | 0.82 | 0.84 | 0.85 | 0.88 | 0.89 | 0.92 |
|  |  |  |  | N | 3169 | 3472 | 3469 | 3474 | 3469 | 3468 | 3471 | 3472 | 3474 | 3471 | 3469 |
| SBO (2003.01–2012.12) | 47.05 | 12.96 | 3108.9 | Bias | −3.09 | −2.04 | −2.26 | −1.66 | −0.68 | 0.28 | 0.77 | −0.43 | −0.76 | −1.67 | −4.34 |
|  |  |  |  | MAB | 8.19 | 7.24 | 6.82 | 6.27 | 6.38 | 6.71 | 7.07 | 7.22 | 7.41 | 7.71 | 9.12 |
|  |  |  |  | RMSE | 10.64 | 9.76 | 9.41 | 8.93 | 9.23 | 9.92 | 10.34 | 10.35 | 10.66 | 10.87 | 12.17 |
|  |  |  |  | R | 0.89 | 0.91 | 0.91 | 0.91 | 0.90 | 0.89 | 0.89 | 0.88 | 0.87 | 0.88 | 0.86 |
|  |  |  |  | N | 3757 | 3759 | 3759 | 3758 | 3759 | 3758 | 3759 | 3761 | 3761 | 3762 | 3762 |
| SMS (2006.04–2017.06) | −89.98 | −24.8 | 2800 | Bias | −15.77 | −15.15 | −15.26 | −13.77 | −12.10 | −9.58 | −8.98 | −7.89 | −7.96 | −8.05 | −7.42 |
|  |  |  |  | MAB | 19.60 | 19.07 | 19.73 | 19.54 | 19.27 | 19.16 | 19.44 | 18.76 | 18.11 | 17.80 | 16.64 |
|  |  |  |  | RMSE | 23.91 | 23.22 | 23.56 | 23.54 | 23.25 | 23.13 | 23.44 | 22.89 | 22.11 | 22.01 | 20.80 |
|  |  |  |  | R | 0.68 | 0.67 | 0.64 | 0.60 | 0.58 | 0.53 | 0.50 | 0.49 | 0.50 | 0.48 | 0.49 |
|  |  |  |  | N | 2330 | 2885 | 2882 | 2855 | 2854 | 2868 | 2866 | 2888 | 2917 | 2605 | 1959 |
| SPO (2000.02–2017.03) | 43.73 | −96.62 | 473 | Bias | −4.56 | −5.83 | −6.66 | −7.01 | −6.80 | −6.11 | −5.08 | −3.81 | −2.83 | −1.51 | −0.22 |
|  |  |  |  | MAB | 8.32 | 9.07 | 9.39 | 9.64 | 9.31 | 8.90 | 8.40 | 7.58 | 7.47 | 7.46 | 7.71 |
|  |  |  |  | RMSE | 11.18 | 12.09 | 12.44 | 12.79 | 12.51 | 11.95 | 11.39 | 10.12 | 10.09 | 9.90 | 10.12 |
|  |  |  |  | R | 0.70 | 0.69 | 0.69 | 0.68 | 0.70 | 0.71 | 0.73 | 0.76 | 0.75 | 0.76 | 0.75 |
|  |  |  |  | N | 2800 | 2794 | 2771 | 2762 | 2719 | 2733 | 2752 | 2756 | 2773 | 2807 | 2824 |

**Table A1.** *Cont.*

| Site | Lat | Long | El | Statistical Parameters | 7:00 | 8:00 | 9:00 | 10:00 | 11:00 | 12:00 | 13:00 | 14:00 | 15:00 | 16:00 | 17:00 |
|------|-----|------|-----|----------------------|------|------|------|-------|-------|-------|-------|-------|-------|-------|-------|
| SXF (2009.01–2019.11) | −69.01 | 39.59 | 18 | Bias | 3.39 | 2.54 | 0.75 | 1.28 | 1.36 | 1.35 | 1.45 | 1.25 | 1.35 | 2.73 | 2.54 |
|  |  |  |  | MAB | 8.53 | 8.74 | 8.24 | 7.63 | 7.48 | 7.38 | 7.46 | 7.82 | 8.29 | 8.69 | 8.69 |
|  |  |  |  | RMSE | 11.56 | 12.37 | 12.08 | 11.14 | 11.02 | 10.77 | 10.95 | 11.38 | 11.90 | 11.93 | 11.80 |
|  |  |  |  | R | 0.86 | 0.85 | 0.85 | 0.86 | 0.86 | 0.86 | 0.86 | 0.85 | 0.85 | 0.85 | 0.83 |
|  |  |  |  | N | 4786 | 5916 | 5932 | 5933 | 5942 | 5946 | 5943 | 5936 | 5940 | 5938 | 4538 |
| SYO (2000.03–2021.04) | 22.79 | 5.53 | 1385 | Bias | 6.69 | 5.49 | 4.67 | 4.28 | 4.73 | 5.29 | 5.64 | 5.84 | 6.88 | 8.52 | 9.45 |
|  |  |  |  | MAB | 13.64 | 12.90 | 12.58 | 12.61 | 12.57 | 12.22 | 12.21 | 11.99 | 12.46 | 12.58 | 13.16 |
|  |  |  |  | RMSE | 18.31 | 17.71 | 17.34 | 17.26 | 17.13 | 16.76 | 16.78 | 16.68 | 17.21 | 17.30 | 17.59 |
|  |  |  |  | R | 0.60 | 0.62 | 0.64 | 0.65 | 0.66 | 0.69 | 0.69 | 0.69 | 0.68 | 0.70 | 0.69 |
|  |  |  |  | N | 4084 | 4616 | 5065 | 5456 | 5756 | 5910 | 5783 | 5455 | 5068 | 4586 | 4063 |
| TAM (2000.03–2021.06) | 36.06 | 140.1 | 25 | Bias | −4.23 | −3.37 | −3.02 | −2.54 | −2.50 | −2.31 | −0.73 | 0.67 | 1.23 | 0.46 | 0.28 |
|  |  |  |  | MAB | 7.02 | 5.76 | 5.13 | 4.98 | 5.28 | 6.04 | 6.48 | 6.95 | 7.33 | 7.96 | 7.96 |
|  |  |  |  | RMSE | 9.02 | 7.61 | 6.85 | 6.86 | 7.24 | 8.39 | 9.73 | 10.70 | 11.19 | 11.21 | 10.73 |
|  |  |  |  | R | 0.76 | 0.74 | 0.72 | 0.67 | 0.62 | 0.65 | 0.72 | 0.77 | 0.80 | 0.79 | 0.76 |
|  |  |  |  | N | 7298 | 7314 | 7318 | 7322 | 7324 | 7321 | 7326 | 7324 | 7326 | 7323 | 7323 |
| TAT (2000.03–2021.05) | 58.25 | 26.46 | 70 | Bias | −1.54 | 0.13 | 0.59 | 0.89 | 1.02 | 0.48 | −0.14 | −1.15 | −1.99 | −2.88 | −6.33 |
|  |  |  |  | MAB | 8.85 | 7.96 | 7.64 | 7.42 | 7.64 | 8.44 | 8.40 | 8.73 | 8.94 | 9.36 | 11.96 |
|  |  |  |  | RMSE | 11.69 | 10.62 | 10.52 | 10.25 | 10.42 | 11.33 | 11.15 | 11.50 | 11.75 | 12.19 | 16.18 |
|  |  |  |  | R | 0.79 | 0.86 | 0.88 | 0.89 | 0.88 | 0.86 | 0.85 | 0.84 | 0.84 | 0.83 | 0.70 |
|  |  |  |  | N | 6443 | 7725 | 7727 | 7726 | 7726 | 7724 | 7725 | 7724 | 7723 | 7724 | 7246 |
| TOR (2000.03–2020.11) | 39.75 | 117 | 32 | Bias | −1.07 | −1.11 | −1.06 | −0.41 | −0.75 | −0.49 | −0.20 | 0.01 | −0.86 | −1.32 | −2.18 |
|  |  |  |  | MAB | 10.58 | 10.07 | 9.40 | 8.57 | 8.55 | 8.82 | 8.98 | 9.74 | 10.71 | 11.18 | 11.41 |
|  |  |  |  | RMSE | 13.45 | 13.05 | 12.37 | 11.36 | 11.30 | 11.67 | 11.84 | 12.64 | 13.93 | 14.34 | 14.33 |
|  |  |  |  | R | 0.80 | 0.84 | 0.85 | 0.88 | 0.88 | 0.87 | 0.87 | 0.84 | 0.79 | 0.75 | 0.72 |
|  |  |  |  | N | 5263 | 6247 | 7567 | 7567 | 7564 | 7569 | 7569 | 7571 | 6750 | 5657 | 4735 |
| XIA (2005.01–2015.10) | 82.49 | −62.42 | 127 | Bias | −1.60 | 2.07 | 2.34 | 2.27 | 1.60 | −2.43 | −2.01 | −2.23 | −1.61 | −1.21 | −1.65 |
|  |  |  |  | MAB | 8.93 | 7.53 | 7.35 | 6.96 | 6.62 | 6.72 | 6.59 | 6.84 | 7.09 | 6.83 | 6.79 |
|  |  |  |  | RMSE | 11.67 | 10.07 | 10.06 | 9.53 | 9.15 | 9.04 | 8.87 | 8.98 | 9.17 | 8.83 | 8.87 |
|  |  |  |  | R | 0.78 | 0.87 | 0.88 | 0.88 | 0.88 | 0.90 | 0.90 | 0.89 | 0.87 | 0.87 | 0.84 |
|  |  |  |  | N | 3635 | 3650 | 3650 | 3650 | 3651 | 3651 | 3652 | 3653 | 3653 | 3653 | 2784 |

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
