# Peer review of "Diurnal Cycle in Surface Incident Solar Radiation Characterized by CERES Satellite Retrieval"

_remotesensing, doi:10.3390/rs15133217_

Round 1
Reviewer 1 Report
Overview and General Impression
The presented manuscript (MS) is dedicated to the description of the performed in the study evaluation of the Surface incident solar radiation (Rs) data, derived from the Clouds and the Earth’s Radiant Energy System Synoptic (CERES) satellite product. The study is performed on a diurnal scale which, according the author’s to claims, is novel, and the ground-based data from the Baseline Surface Radiation Network (BSRN) is used as an independent reference to evaluate hourly Rs.
Alongside the straightforward and clear presentation of the results, I could outline two other strengths of the work:
- Stations (with suitable record structure) from the whole globe are considered.
- The results are presented separately depending on:
-- the geographical position (continent interior/polar/ coastal and island)
-- the degree of sky cloud coverage (three categories).
In general, the work is well conceptualized and structured. As a result, the MS is relatively well prepared, both in textual and graphical parts. It fits well in the thematic scope of ‘Remote sensing’ and has the potential to attract readers.
The MS suffers, however, from some weaknesses which should be addressed before it becomes publishable.
Major remarks:
My two remarks concern the adopted evaluation strategy.
First, it remains, at least for me, the source of the data for the radiance of the TOA in for the evaluated (i.e. CERES) and reference (BSRN) dataset. It is stated that “... the observational data and CERES retrieved data were standardized as follows…” (r162-164), i.e. it is applied independently to both datasets. I am not familiar with CERES and BSRN but how ground-based data can be a source of TOA-records? If Swi,TOA (the quantity in the denominator in Eq.1) is not the same for both datasets then part of the reported later differences CERS vs. BSRN could be linked to this variable, rather than solely to the inspected ground parameters.
An explanatory paragraph that addresses these questions should be included in the revised version.
Second, why only sign-insensitive statistical scores in the evaluation are considered, for example, MAB? It is stated that “... to avoid the offsetting of positive and negative deviations…” (r179 – r180) but this is not convincing hence additional result for possible/probable (systematic) under/overestimation of CERES in respect to the ‘ground truth’ under some of the considered stations/conditions could be a valuable outcome of the study.
This also should be explained.
Minor remarks:
r40: circulation → cycle
Figures 10-11 appear blurry. I don’t know where the reason is but this should be corrected.
Not least, the study relies entirely on freely available data. Thus acknowledgment to the CERES and BSRN providers should be stated. Use the standard recommended phrases.
The quality of the English language appears acceptable.
Author Response
Dear Editor and Reviewers,
We hereby resubmit the manuscript "Diurnal cycle in surface incident solar radiation characterized by CERES satellite retrieval" by Lu Lu and Qian Ma to Remote Sensing (ISSN 2072-4292).
Following the suggestions from the editor and reviewers, we carefully revised the paper and rewrote the paper. All the comments have been addressed and a point to point response was attached.
Thanks for your consideration of our resubmission. We are looking forward to your evaluation of this paper.
Best regards,
Lu Lu and Qian Ma
Attachment: Response to editor and reviewers ' comments

Reviewer 2 Report
The paper “Diurnal cycle in surface incident solar radiation characterized by CERES satellite retrieval” compare hourly solar radiation (Rs) retrieved from CERES satellite product on the diurnal scale, with Rs observed from 53 stations belonging to the Baseline Surface Radiation Network (BSRN). The study is carried out using data from 2000 to 2021.
The draft is in general well, with the need to improve the quality of figs 6 to 10 and include the missing figure 12. Also conclusions are not presented.
Figure 10 needs to be corrected/explained, because the statistical parameters present a behavior opposite to that presented in Figure 9. In Figure 9 it can be seen that when the values ​​of MAB and RMSE are high, those of R decrease and vice versa, for the different times and groups of sites (continental, island/coastal, polar and total) analyzed. On the contrary, in figure 10 the statistics corresponding to clear sky conditions present the lowest values ​​of MAE and RMSE but the values ​​of R are intermediate. Something similar occurs if the daily variation of these statistics is analyzed for this sky condition, the minimum MAB, RMSE and R values occurs near noon.
Minor comments are highlighted in the draft.

Author Response

(The authors gave the same response as above.)

Reviewer 3 Report
The authors used independently widely distributed ground-based data from the BSRN to evaluate hourly Rs from the CERES Hour product, and explored the influence of cloud cover and aerosols on the diurnal variation in Rs. They found that CERES-retrieved Rs performs better at midday (~8.80% for MAB, ~12.50% for RMSE, and ~0.84 for R) than at sunrise and sunset (~10.00% for MAB, ~13.50% for RMSE, and ~0.80 for R). For spatial distribution, CERES-retrieved Rs performs better over the continent than over the island/coast and polar regions. The change in bias in Rs caused by cloud cover is 1.97%-5.38% larger than 0.31%-2.52% by AOD. The comments and suggestions are in the following.
Abstract, should be shortened.
Figure 1,There are 8 blue points, but in the main test, it says 7 in the polar region.
Line 129, “We focus on the period from 7:00 to 17:00 each day”, It may loss about 50% data in the evaluation for the period not in 7:00-17:00. Thus, it is suggested to also evaluate Rs during these time period for a full understanding of CERES data.
Line 183, Most Rs data are in the range from 0.5 to 0.75, please give the unit in the main text and related figures (4, 5, etc.), explain how many station’s data, sample size, etc. were used.
Lines 200-201, R is higher than 0.8 most, R is higher, R should be Rs?
Figures 9, 10, 11 are not clear.
Line 233, Where is Table 1?
It is suggested to show more detailed information for each station in a table, including time period, sample number, standard deviations of CERES and ground observations, correlation, MAB, RMSE, etc.
Lines 268-270, please check the numbers in “under cloudy-sky (9.96%-11.91% for MAB and 13.42%-15.20 for RMSE) and overcast-sky conditions (10.32%-11.09% for MAB”.
Lines 326-329, please give the evidence for the diurnal variation in biases in CERES etrieved Rs is more sensitive to cloud cover than AOD.
Lines 334-346, The impact of diurnal and seasonal variation on the Rs evaluation was removed by dividing the solar radiation on the TOA. It may be not right, as the solar radiation on the TOA is a constant.
A conclusion section is suggested.
Author Response

(The authors gave the same response as above.)

Reviewer 4 Report
The author conducted a study on the observed Rs data from different sites in the BSRN observational network. The purpose of this study was to evaluate CERES-retrieved hourly Rs data and investigate the influence of cloud cover and AOD on the bias of the CERES-retrieved Rs data. Although the subject is important, there are several areas where the paper can be improved. Here are some remarks:
-
The figures in the paper lack clarity. It is suggested to enhance the visibility of the figures, possibly by using colors to make the information more comprehensible( eg. Figure 2).
-
Figure 12 is missing in the section discussing AOD. Please include Figure 12 or clarify its absence.
-
More details about the CERES-retrieved hourly Rs data and the methodology employed should be provided. This will help readers understand the data source and the techniques used for retrieval.
-
There are numerous references in the paper. It is recommended to only retain the most relevant and important ones. This will streamline the paper and make it more focused.
-
On page 13, the statement "Aerosols were divided..." requires clarification. Is it referring to the AOD or some other aspect related to aerosols? Additionally, it would be helpful to specify which spectral band was utilized in this context.
Overall, by addressing these remarks, the paper can be significantly improved in terms of figure clarity, the inclusion of missing figures, providing more detail on methodology and CERES-retrieved data, streamlining the bibliography, and clarifying statements related to aerosols.
Author Response

(The authors gave the same response as above.)

Round 2
Reviewer 2 Report
The authors corrected/completed most of the suggestions, however there are still minor issues that are highlighted in the draft. I consider that it is not necessary to revise again the work.

Author Response
Dear Editor and Reviewers,
We hereby resubmit the manuscript "Diurnal cycle in surface incident solar radiation characterized by CERES satellite retrieval" by Lu Lu and Qian Ma to Remote Sensing (ISSN 2072-4292).
We carefully considered all the reviewer's comments and made corresponding modifications to the manuscript. A point to point response to reviewer's comments was attached.
Thanks for your consideration of our resubmission. We are looking forward to your evaluation of this paper.
Best regards,
Lu Lu and Qian Ma

Reviewer 4 Report
No comments
Author Response

(The authors gave the same response as above.)
